# Linear and nonlinear chromatic integration in the mouse retina

Mohammad Hossein Khani 🔟 [1,2,3 ✉] & Tim Gollisch 🔟 [1,2 ✉]

The computations performed by a neural circuit depend on how it integrates its input signals into an output of its own. In the retina, ganglion cells integrate visual information over time, space, and chromatic channels. Unlike the former two, chromatic integration is largely unexplored. Analogous to classical studies of spatial integration, we here study chromatic integration in mouse retina by identifying chromatic stimuli for which activation from the green or UV color channel is maximally balanced by deactivation through the other color channel. This reveals nonlinear chromatic integration in subsets of On, Off, and On–Off ganglion cells. Unlike the latter two, nonlinear On cells display response suppression rather than activation under balanced chromatic stimulation. Furthermore, nonlinear chromatic integration occurs independently of nonlinear spatial integration, depends on contributions from the rod pathway and on surround inhibition, and may provide information about chromatic boundaries, such as the skyline in natural scenes.

[1] Department of Ophthalmology, University Medical Center Göttingen, Göttingen, Germany. [2] Bernstein Center for Computational Neuroscience, Göttingen, Germany. [3] International Max Planck Research School for Neuroscience, Göttingen, Germany. ✉email: mhkhanni@gmail.com; tim.gollisch@med.uni-goettingen.de

Neurons in sensory systems encode different features of the environment by integrating signals that are detected in receptor cells. In the visual system, the signal detection occurs in the photoreceptors, which are distributed over space on the retina and which belong to different chromatic channels, depending on their spectral sensitivity. Downstream neurons, therefore, integrate visual information over time and space as well as over color channels, and the signal transformations inherent in these integration processes shape the computation and feature extraction associated with a given neuron. Temporal and spatial signal integration have been intensively studied in the retina, where they have been linked to phenomena such as temporal filtering, adaptation, motion detection, and other specific visual functions[1–9]. Studies of spatial integration, in particular, refined the idea of receptive fields[10,11] and later resulted in the distinction of linear and nonlinear spatial integration, as originally exemplified by the X and Y cells of the cat retina[12–14]. Mechanistic investigations of spatial integration then elucidated the role of retinal bipolar cells in shaping signal transmission through the retina[15–18] and helped characterize the suppressive receptive field surround[11,19–21].

Unlike the temporal and spatial aspects of retinal signal integration, however, the properties of chromatic integration are largely unknown and direct assessments are lacking. Chromatic signal processing starts at the level of photoreceptors where visual signals are separated into different input channels, based on the wavelength selectivity of the photoreceptors' opsins. The chromatic signals are transferred to bipolar cells, which can sample the signals from different photoreceptor types either selectively or non-selectively, potentially with some bias towards one photoreceptor type[22–25]. Little is known about how these chromatically selective and non-selective signals are integrated by the ganglion cells.

So far, chromatic signal processing has been mostly connected with color-opponent cells, which constitute a specific subpopulation of ganglion cells[26–33]. But other types of ganglion cells also combine signals from different chromatic channels and could do so in different ways, for example, linearly or nonlinearly. Thus, a thorough understanding of retinal signal integration requires including chromatic integration beyond the studies of color-opponent cells and approaching it similarly to temporal and spatial integration.

In the present study, we demonstrate how chromatic integration can be investigated in a way that is conceptually analogous to the classical studies of spatial integration[12–14]. We measured the linearity and nonlinearity of chromatic integration in the mouse retina through visual stimuli that simultaneously changed the activation of different chromatic channels in opposite directions and searched for either response cancellation or robust responses to both reversal directions of this contrast combination. Responses of retinal ganglion cells recorded with multielectrode arrays revealed both linear and nonlinear chromatic integration in different cells, independently of their linear or nonlinear spatial integration. We further found that nonlinear On and nonlinear Off cells differ systematically in how the nonlinearities affect their responses; under stimulation that balances the contributions from different chromatic channels, nonlinear Off cells (as well as nonlinear On–Off cells) are activated, whereas nonlinear On cells are suppressed. Finally, investigations with localized stimulation, varying light level, and pharmacological intervention indicated that contributions from the rod pathway together with inhibitory signaling in the receptive field surround create the nonlinearity of chromatic integration in the retina.

## Results

**Color as a dimension of retinal signal integration.** Retinal neurons are often functionally characterized by how they integrate input signals over time and space[2,14,34]. Yet, in addition, retinal neurons also integrate signals chromatically. Photoreceptors separate light into pathways that represent different wavelengths (colors), and downstream neurons pool signals from these chromatic channels. The photoreceptors in the mouse retina, for example, have peak sensitivities in the UV (S-cones) and green range (M-cones and rods). Retinal ganglion cells can therefore combine signals from UV and green light, and this can occur in a linear or nonlinear fashion, depending on the characteristics of signal transmission in the corresponding retinal circuits. To study this chromatic integration, we drew analogies to classical studies of spatial integration[12–14]. These studies investigated the responses of retinal ganglion cells under contrast-reversing spatial gratings (or presentation and withdrawal of the grating) and searched for nulling of responses, that is, no evoked activity for either reversal direction. Response nulling indicated a cancellation of the activation from luminance increases and decreases, which is a sign of linear spatial integration, and the corresponding cells were called X cells. So-called Y cells, on the other hand, displayed increased activity for both reversal directions, taken as a sign for a lack of cancellation and thus nonlinear integration.

Figure 1 shows that such different response characteristics also occur for chromatic integration. The two displayed sample cells from the mouse retina were both Off cells and responded with bursts of spikes when either the green or the UV illumination was decreased (negative contrast; Fig. 1a, b). For a particular contrast combination, however, with decreased illumination of one color and increased illumination of the other (Fig. 1c), Cell 1 remained silent regardless of whether green contrast was positive and UV contrast was negative or whether the contrast-reversed version was applied. Thus, akin to the X cells of spatial integration, positive and negative activation in the two chromatic channels can cancel each other, providing evidence of linear chromatic integration by this cell. Cell 2, on the other hand, displayed vigorous spiking for both of these opposing contrast combinations. Analogous to Y cells, this cell was therefore activated by contrast-reversed stimuli without cancellation, indicating nonlinear chromatic integration.

**A chromatic stimulus to identify the balance point of two color channels.** Identifying linear and nonlinear chromatic integration as in Fig. 1 relies on identifying the right contrast combination where the two color channels balance each other, leading either to response cancellation or to equally strong responses for a stimulus and its contrast-reversed version. To systematically search for this chromatic balance point, we designed a chromatic-integration stimulus with different chromatic contrast combinations of opposite signs that spanned the range from pure green to pure UV stimulation (Fig. 2a, b). Concretely, we stimulated the retina with two sets of full-field UV/green stimuli, containing eleven combinations of green-On-UV-Off and the eleven contrast-reversed combinations, respectively (Fig. 2b). Within each of the two stimulus sets, the negative contrast of one color channel decreased from −20 to 0% (in steps of 2%) while the positive contrast of the other color channel increased from 0 to 20%. This is analogous to the scenario of spatial integration measured with a spatial grating where a black/white boundary is shifted across the receptive field to change the balance of the areas with negative and positive contrast. Based on these 22 chromatic contrast combinations, we searched for a balance point, which yielded the same response of a ganglion cell for a green-On-UV-Off stimulus and its contrast-reversed version.

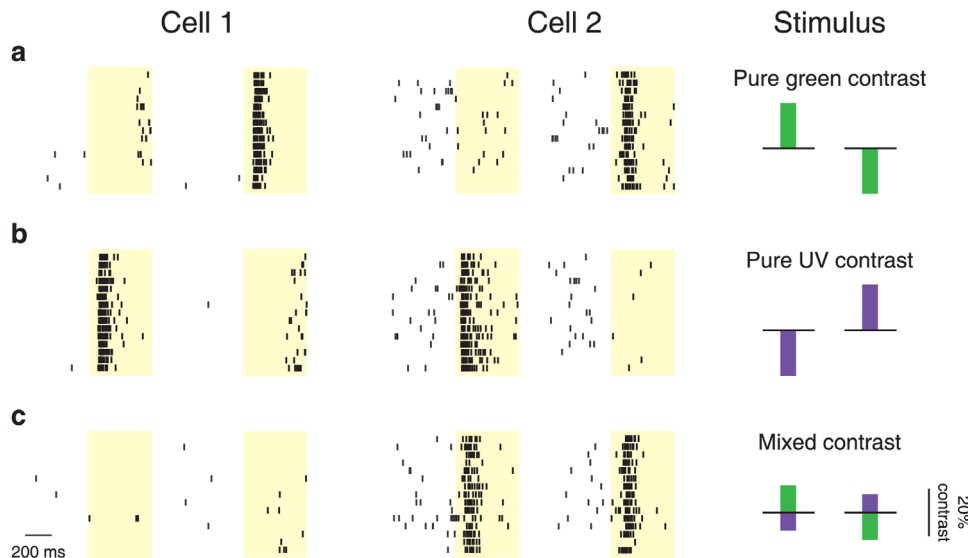

**Fig. 1 Linear and nonlinear chromatic integration in the mouse retina. a** Raster plots of spike times for two sample Off-type ganglion cells in response to 500-ms presentations (marked by the yellow shaded areas) of green contrast. Each line is a trial, showing for each cell spikes from 500 ms prior to stimulus onset until stimulus offset for both positive contrast (left) and negative contrast (right). Schematic presentations of the chromatic stimulus components are shown in the column on the right. **b** Responses of the same cells to pure UV stimuli with negative contrast (left) and positive contrast (right). **c** Responses of the same cells to a stimulus with simultaneous UV and green contrast of opposing signs (mixing the pure-color stimuli from **a** and **b** in the same column above) and to the contrast-reversed stimulus. Cell 1 showed cancellation of activation by green and UV contrast (linear chromatic integration), whereas Cell 2 displayed strong activity for both contrast combinations (nonlinear chromatic integration).

Responses of mouse retinal ganglion cells were recorded with multielectrode arrays, and stimuli were delivered via a projection system with two LEDs suited for activating mouse photoreceptors (Supplementary Fig. 1). To better separate signals coming from the two cone pathways of the mouse retina, originating in the S- and M-opsins of cone photoreceptors, we used the method of silent substitution[35] to present opsin-isolating stimuli. Note, though, that the relative spectral sensitivity of the mouse M-opsin is nearly identical to that of the mouse rod opsin[36–38]. Rods will therefore experience similar effective contrast as the M-opsins and, in particular, will not be activated by S-opsin-isolating stimuli. All color contrast values in this work, therefore, imply contrast on the level of opsin activation, with UV contrast standing for S-opsin activation and green contrast for M-opsin and rod activation.

The 22 contrast combinations were displayed repeatedly in random order for 500 ms each, separated by 2 s of background illumination. Responses were measured as firing rates after stimulus onset, with background firing rate subtracted (Fig. 2c). For each cell, we then compared responses for the pairs of contrast-reversed chromatic stimuli (Fig. 2d) and plotted the extracted firing rates together so that each color combination and its reversed version are aligned on the same x axis, as shown in Fig. 2e and f for two sample cells. In the following, we refer to these displays as chromatic-integration curves.

The chromatic-integration curves allow easy identification of the balance point as the crossing point of the two curves. At this point, one chromatic stimulus combination and its contrast-reversed version induce the same response, indicating balanced input from the two chromatic channels. Whether the response level at the crossing point is near zero or deviates from zero is therefore indicative of whether chromatic integration is linear (Fig. 2e) or nonlinear (Fig. 2f), respectively.

**Distinct patterns of nonlinear chromatic integration in On and Off cells.** We recorded the responses of around 3,400 mouse

retinal ganglion cells to the chromatic-integration stimulus. Figure 3a–c shows responses of six different ganglion cells to this stimulus, representative of the range of response patterns that we observed. The first three examples are Off, On, and On–Off cells that all displayed linear chromatic integration as evident from the crossing points of the color integration curves near zero (Fig. 3a). This is reflected in the corresponding PSTHs, where no difference from the background activity is apparent (Fig. 3d). The other three sample cells are Off, On, and On–Off cells that showed nonlinear chromatic integration, as their crossing points were offset from zero (Fig. 3b). The nonlinear Off and On–Off cells displayed increased firing rates at the crossing point (Fig. 3e), similar to the response of Y cells to reversing gratings. The nonlinear On cell, on the other hand, showed a different response type with suppressed activity below the baseline at the crossing point (Fig. 3f).

Besides the linear and nonlinear cells, we found ganglion cells that responded only to UV light ($N = 319$; about 10% of recorded cells; see Supplementary Fig. 2a–c for examples), to which we refer as UV-selective cells. Relatedly, previous studies had shown particularly high sensitivity to UV light for some mouse ganglion cells[39,40], although not exclusive selectivity. Furthermore, we found a small population ($N = 52$; about 2%) of color-opponent cells (see Supplementary Fig. 2b-c for examples). Previous reports with full-field stimulation had reported similar percentages of cells with chromatic opponency in mouse retina[26,41] and mouse LGN[42]. For these cells, one chromatic-integration curve always lay above the other without any intersection because this chromatic-integration curve corresponded to the preferred opponency (e.g., green-On-UV-Off). For the present study, UV-selective and color-opponent cells were not the focus, and their properties were therefore not analyzed further.

To quantify the degree of nonlinearity in chromatic integration, we defined a chromatic nonlinearity index, calculated as an estimated lower bound on the actual response at the crossing point, normalized by the maximum overall response of the cell

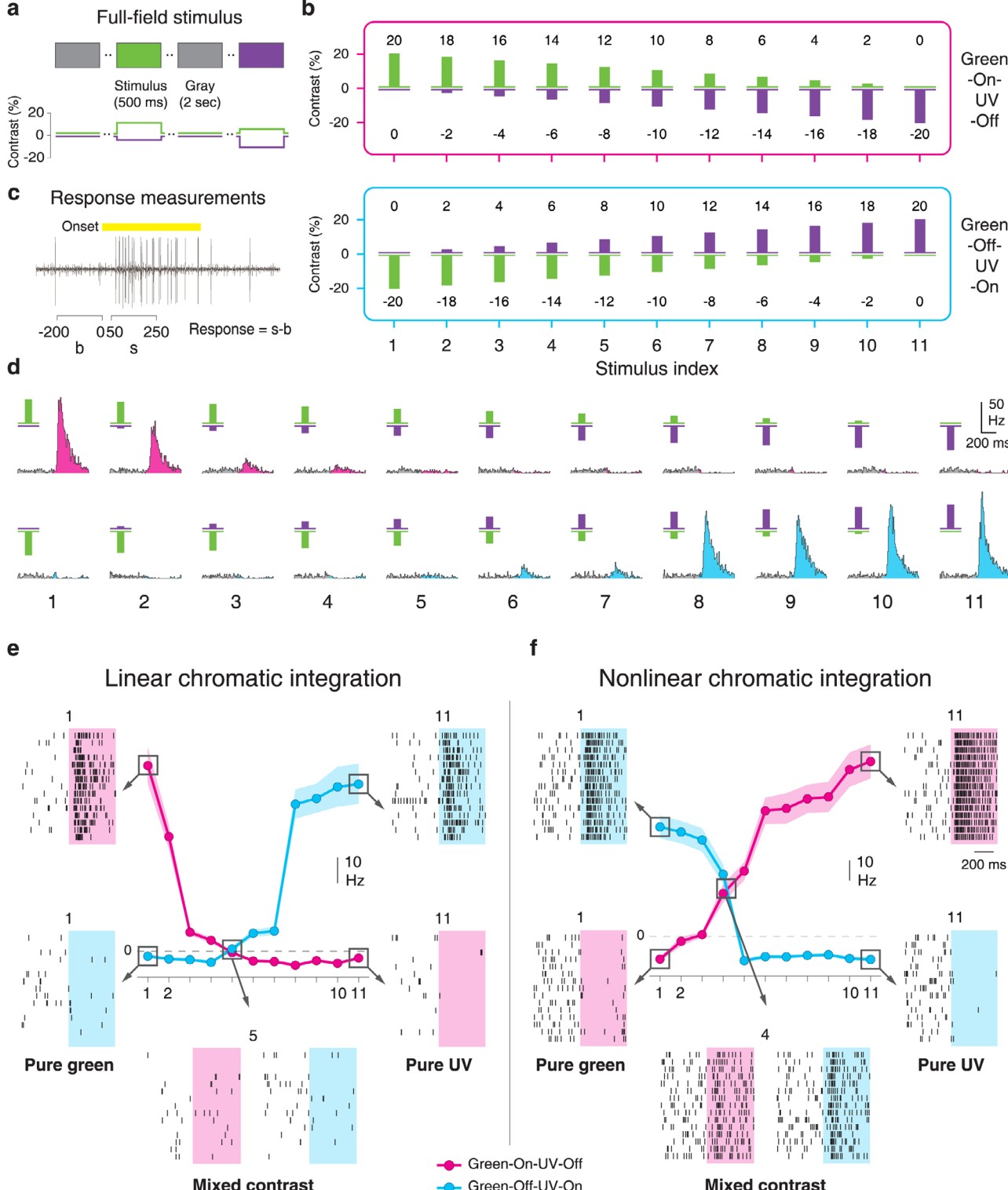

**Fig. 2 Measurement of chromatic integration. a** Schematic representation of the chromatic-integration stimulus. 500-ms presentations of opposing contrast of UV and green were interleaved with 2 s of background illumination (gray). **b** Schematic depictions of all contrast combinations used for assessing chromatic integration. The contrast combinations are arranged into two sets, corresponding to green-On-UV-Off (magenta box, top) and to green-Off-UV-On (blue box), with pairs of contrast-reversed combinations vertically aligned and assigned to a stimulus index (1–11). **c** Response measurements used for constructing the chromatic-integration curves. Spike counts (s) were obtained over a 200-ms window during stimulation (50–250 ms after stimulus onset), and the background activity (b), measured as the spike count during the 200 ms prior to the stimulus, was subtracted (s − b). The yellow bar shows the stimulus period. **d** PSTHs for a sample On cell responding to the contrast combinations of the chromatic-integration stimulus (below: stimulus indices 1–11). **e** Chromatic-integration curves for the same cell as in **d**, surrounded by exemplary raster plots for some of the contrast combinations. At the balance point, activity is canceled out, which indicates linear chromatic integration. **f** Same as **e**, but for a sample Off cell with sizeable responses at the balance point, which indicates nonlinear chromatic integration. Shaded regions around the data curves mark the range of mean ± SEM.

(see "Methods" section). The index takes values close to zero if activity stays near baseline at the crossing point, whereas strong responses at the crossing point lead to positive index values and negative indices imply response suppression. We found that the nonlinearity indices of all recorded cells were distributed in a continuous fashion (Supplementary Fig. 2d), with a strong peak around zero, but extended tails in both directions, in particular towards positive index values. Thus, the majority of cells have approximately linear chromatic integration, but at both ends of this continuum, cells with pronounced nonlinear integration can be observed, identified by clear responses at the crossing point. For further analyses, we selected cells as nonlinear by applying a threshold of ±0.1 on the chromatic nonlinearity index (Supplementary Fig. 2d, shaded region), corresponding to a change in firing rate at the crossing point of at least 10% of the maximum response. We chose this threshold ad-hoc and based on the distribution of the nonlinearity indices (Supplementary Fig. 2d). Note that this separation of linear and nonlinear integration is not intended to define specific types of ganglion cells, but rather aimed at simplifying subsequent analyses of the properties of nonlinear chromatic integration. Based on our defined threshold of nonlinearity indices, about one-third of recorded cells showed nonlinear chromatic integration (Fig. 3g). Note, though, that these numbers may not represent the true distributions of these cell categories in the retina because multielectrode-array recordings likely sample the different ganglion cell types in an uneven fashion.

Analyzing the population of all recorded cells showed that the difference of having decreased versus increased activity at the crossing point, as apparent in the examples of Fig. 3b, was a systematic difference between nonlinear On cells on the one hand and nonlinear Off as well as On–Off cells on the other hand (cf. Fig. 3e, f). To assess this, we distinguished On, Off, and On–Off cells by comparing the responses to positive and negative contrast steps in one of the two color components and defining an On–Off index (see "Methods" section). The identified Off and On–Off cells had distributions of nonlinearity indices with extended tails towards positive values, but not negative values, whereas On cells displayed a tail only towards negative values (Fig. 3h). This means that all nonlinear On cells in our recordings showed activity suppression for balanced UV and green stimulation with opposing contrast, whereas all nonlinear Off and nonlinear On–Off cells had increased activity at this balance point. Suppressed activity as observed for nonlinear On cells requires at least some level of baseline activity, and indeed nonlinear On cells displayed a significantly higher level of spontaneous activity than linear cells and other classes of nonlinear cells (Supplementary Fig. 2e).

Given that nonlinear On–Off cells displayed a response peak at the crossing point like Off cells, but not response suppression as seen for On cells, we examined whether the nonlinearity of the On–Off cells is driven by their Off or On responses. Consider, for example, the chromatic-integration curves of the nonlinear On–Off cell in Fig. 3b. At the crossing point, the green-Off-UV-On curve (dark green) has a negative slope, which indicates that responses here are governed by the decreasing green-Off contrast towards the right, not by the increasing UV-On contrast. Analogously, the green-On-UV-Off curve (light green) also suggests that Off responses are dominant at the crossing point, as the curve has a positive slope here and the UV-Off contrast increases towards the right, indicating that responses are driven by the increasing UV-Off contrast. Comparing the slopes of the two chromatic-integration curves systematically by computing a "balance point polarity bias" as the difference of the slopes at the balance point (see "Methods" section and Supplementary Fig. 3a), we found that the balanced point is biased towards Off responses

for most nonlinear On–Off cells (Supplementary Fig. 3b). This suggests that the nonlinearity of nonlinear On–Off cells occurs through the Off responses of the cells. Thus, the same mechanism might be responsible for the increase in firing rate at the balance point in both nonlinear Off and nonlinear On–Off cells.

**Chromatic integration is independent of spatial integration.** Previous studies of spatial integration had shown that many ganglion cells display spatial nonlinearities[14,15,43–45]. Thus, one may ask whether some ganglion cells are generally nonlinear for chromatic integration as well as spatial integration whereas others are generally linear. To test this hypothesis and study the relationship between spatial and chromatic integration, we applied achromatic reversing gratings as typically used to study spatial integration. Figure 4a–d shows the reversing-grating responses of two chromatically linear and two chromatically nonlinear Off cells. For both of these chromatic groups, we observed spatially linear cells (Fig. 4a, c) as well as spatially nonlinear cells (Fig. 4b, d), as identified by their responses to both reversal directions of high-frequency gratings.

To quantify the degree of spatial nonlinearity for all cells, we computed a spatial nonlinearity index from the relative strength of the frequency doubling in the responses to reversing gratings (see "Methods" section). Plotting the spatial nonlinearity index against the chromatic nonlinearity index of all recorded cells (Fig. 4e) showed no clear relation between the two indices, despite a weak, yet significant positive correlation ($r = 0.15$, $p = 0.018$). For example, among chromatically nonlinear On and Off as well as chromatically linear cells, we found spatially linear as well as nonlinear cells with similar distributions of spatial nonlinearity indices (Fig. 4f). The On–Off cells were excluded from this analysis due to ambiguity between their On–Off characteristics and their linear or nonlinear spatial integration properties. Thus, spatial and chromatic nonlinear integration are distinct phenomena that occur for largely independent subsets of ganglion cells. This led us to hypothesize that the mechanism for nonlinear chromatic integration is different from the one that underlies nonlinear spatial integration, which results from the nonlinear pooling of excitatory bipolar cell inputs within the ganglion cell's receptive field center[15,17,45].

**Nonlinear chromatic integration is reduced under grating stimulation.** Spatial integration is traditionally not only analyzed with reversing gratings but also with drifting gratings[13,45,46]. Analogously, we aimed at probing chromatic integration with slowly drifting gratings (1 Hz temporal frequency) for comparison with the results from spatially homogeneous color combinations. The spatial grating was composed of a sinusoidal UV component and a sinusoidal green component of the same spatial frequency, but phase-shifted by 180°, so that maximum UV illumination aligned with minimum green illumination and vice versa (Fig. 5a). Similar to measurements with spatially homogeneous stimuli, we searched for a balance point by using different contrast combinations (Fig. 5b), ranging for each color from 0 to 20% (defined as the relative deviation of the peak amplitude of the sinusoid to the mean). The spatial period of the grating was set to 480 μm so that a half period was larger than most receptive field centers of mouse ganglion cells[47]. This avoided nonlinear effects of spatial integration, which influence responses at higher spatial frequencies[14,48]. For chromatically linear cells, we expected that there should again be a balance point that nullifies responses, whereas chromatically nonlinear cells may show frequency doubling at the balance point because preferred contrast of either color channel should modulate the firing rate without cancellation by the other color channel.

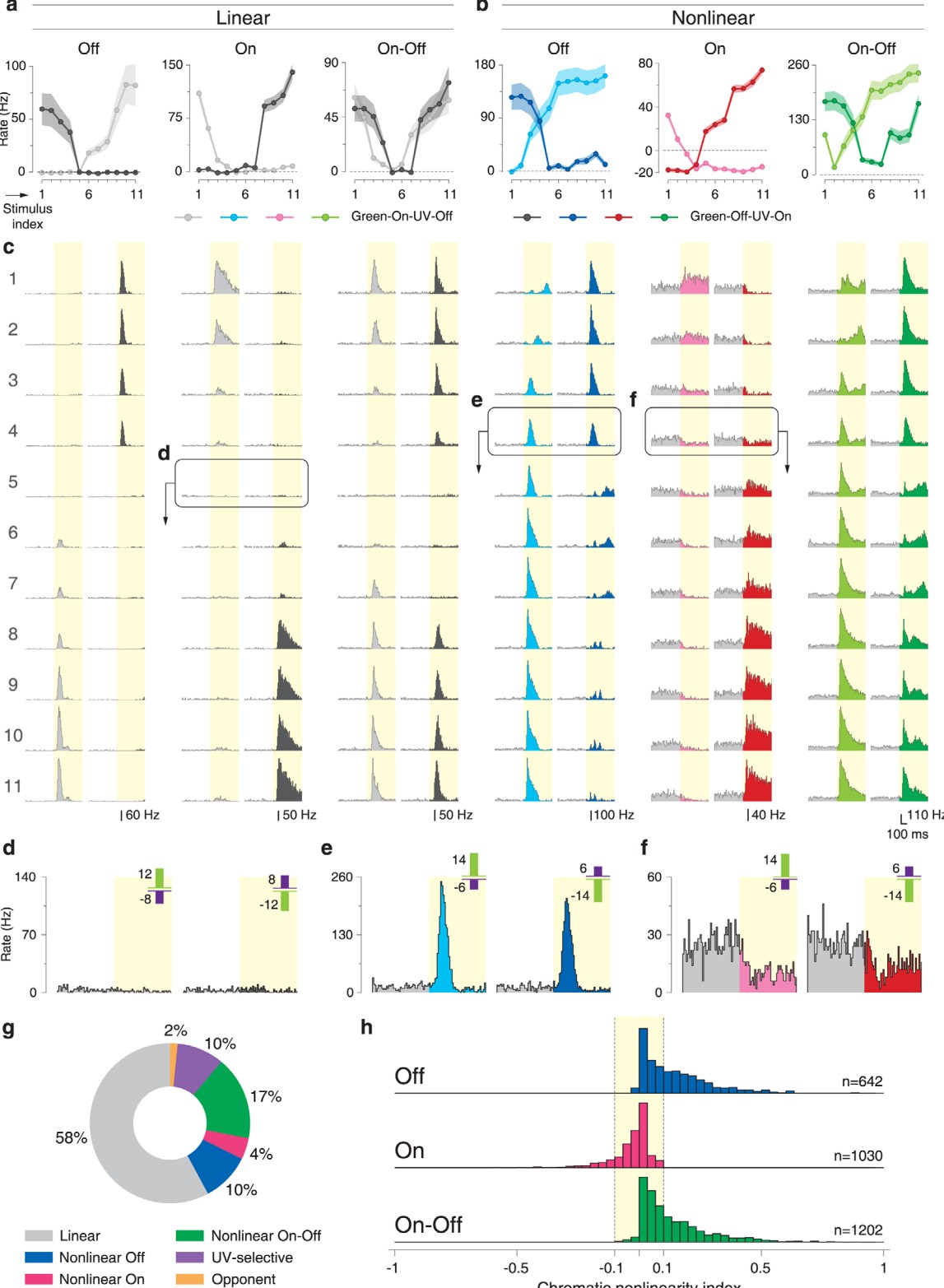

**Fig. 3 Different patterns of nonlinear chromatic integration in On and Off cells. a** Responses of sample linear Off, On, and On–Off cells to the chromatic-integration stimulus. Shaded regions around the curves show mean ± SEM. **b** Reponses of sample nonlinear Off, On, and On–Off cells to the chromatic-integration stimulus. Note that the nonlinear On cell, unlike the other two cells, showed activity suppression at the balance point. Shaded regions around the curves show mean ± SEM. **c** PSTHs for the six sample cells in **a** and **b** for all contrast combinations of the chromatic-integration stimulus. **d–f** PSTHs in response to stimuli near the balance point for a sample linear cell (**d**), the sample nonlinear Off cell (**e**), and the sample nonlinear On cell (**f**). **g** Relative proportions of linear and nonlinear cells among all the recorded ganglion cells. **h** Distributions of chromatic nonlinearity indices for Off, On, and On–Off cells.

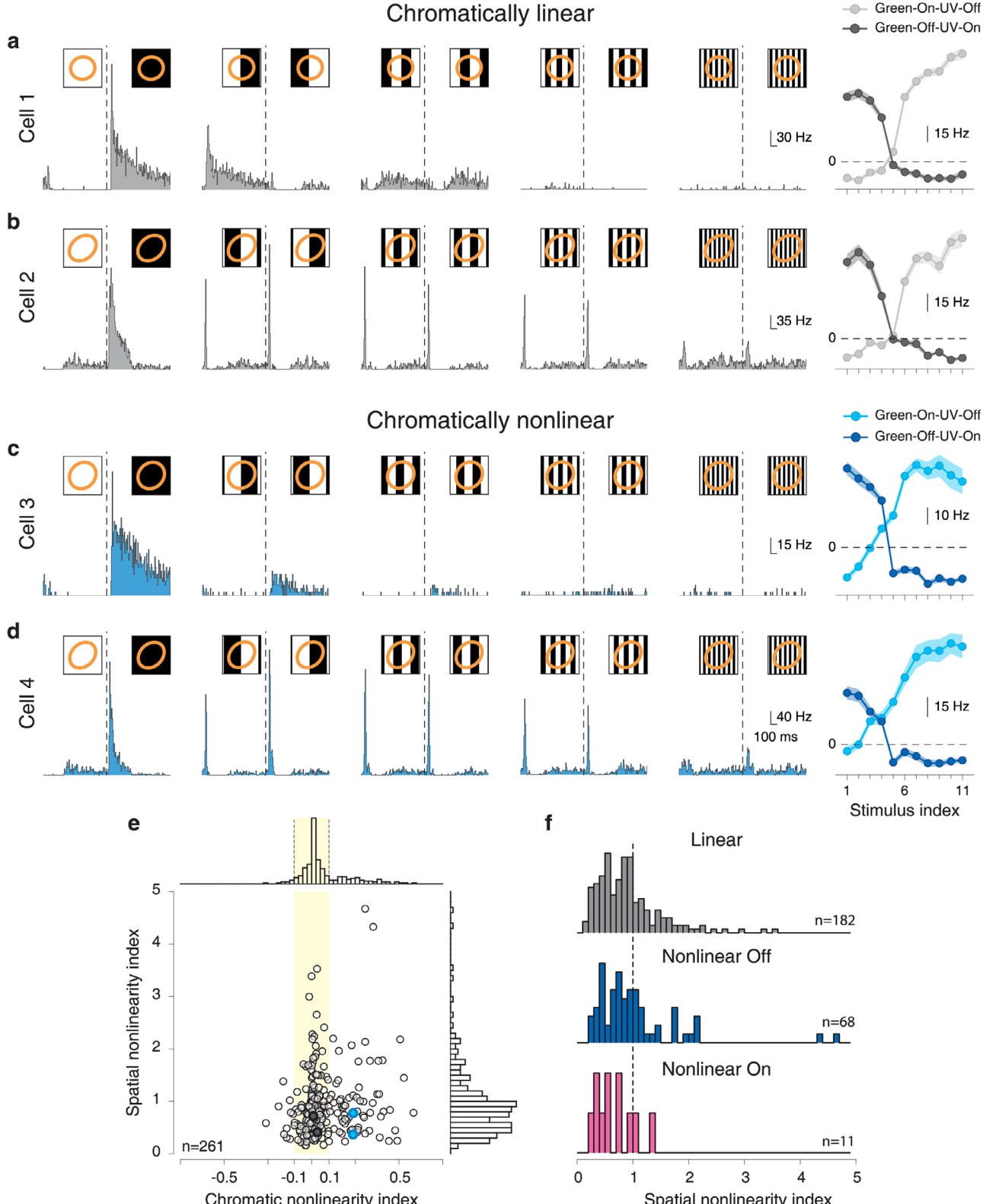

**Fig. 4 Chromatic integration is independent of spatial integration. a** Responses of a chromatically linear cell to reversing-grating stimuli of different spatial scales, showing linear spatial integration. The dashed lines mark the time of the grating reversal. The insets display the stimulus presented to the cell, together with the 1.5-$\sigma$ contour of the Gaussian fit to the receptive field (orange ellipse). To the right: the cell's chromatic-integration curves. **b** Same as **a**, but for a chromatically linear cell that shows nonlinear spatial integration. **c** Same as **a**, but for a chromatically nonlinear cell that shows linear spatial integration. **d** Same as **a**, but for a chromatically nonlinear cell that shows nonlinear spatial integration. The shaded region around the chromatic-integration curves in **a–d** show mean ± SEM. **e** Spatial nonlinearity indices of all On and Off cells versus their chromatic nonlinearity indices, displaying only weak correlation (Pearson correlation coefficient $r = 0.15$, $p = 0.018$, $n = 261$). Sample cells from **a–d** are marked by filled circles of the corresponding color. **f** Distribution of spatial nonlinearity indices for chromatically linear as well as for chromatically nonlinear Off and On cells.

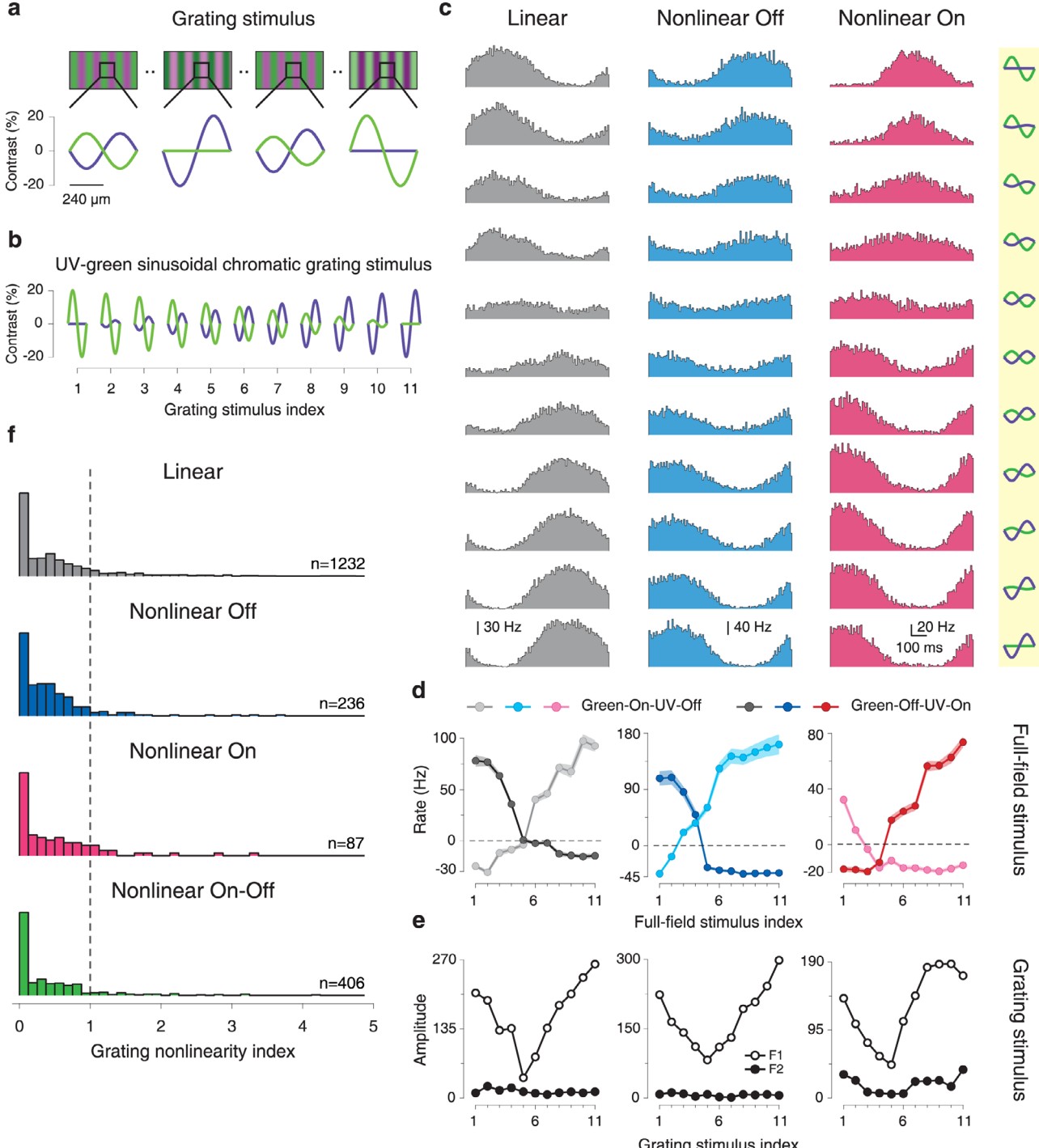

**Fig. 5 Nonlinear chromatic integration is reduced under grating stimulation. a** Four sample gratings of the chromatic grating stimulus, showing the green and UV sinusoids in anti-phase. Below each grating, the contrast over a spatial period is shown for each stimulus frame. **b** Schematic view of all contrast combinations used for the chromatic grating stimulus, displayed over one spatial period. **c** PSTHs over one temporal period for a linear, a nonlinear Off, and a nonlinear On cell in response to the chromatic grating stimulus. The corresponding contrast combinations are depicted to the right. **d** Chromatic-integration curves of the cells shown in **c**, measured with the spatially homogeneous chromatic-integration stimulus. Shaded regions around the curves show mean ± SEM. **e** Amplitudes of first (F1) and second (F2) harmonic of the sample cells, obtained from the PSTHs shown in **c**. **f** Distribution of the grating nonlinearity index for chromatically linear ($n = 1232$) as well as nonlinear Off ($n = 236$), nonlinear On ($n = 87$), and nonlinear On–Off ($n = 406$) cells, with the classification obtained from the spatially homogeneous chromatic-integration stimulus.

To analyze the responses of a ganglion cell to the chromatic drifting gratings, we computed a PSTH over one temporal period for each contrast combination. Figure 5c shows PSTHs of three sample cells, which were classified as chromatically linear, nonlinear On, and nonlinear Off, according to the analysis of

the chromatic-integration curves (Fig. 5d, see Supplementary Fig. 4a–c for examples of On–Off cells). Surprisingly, the PSTHs under the drifting gratings displayed quite a similar structure for all three cells, with strongly reduced modulations of firing rate at mixed green/UV contrast and no indication of frequency

doubling. Fourier analysis of the PSTHs confirmed this. For all three sample cells, the first harmonic, corresponding to the stimulus frequency, was always larger than the second harmonic (Fig. 5e), thus providing no indication of nonlinear spectral integration even for the two cells that had clear responses at the balance point under spatially homogeneous stimulation (Fig. 5d).

To quantify the occurrence of frequency doubling for each recorded cell as a sign of potential nonlinear chromatic integration under grating stimulation, we defined a grating nonlinearity index. Analogous to indices applied to analyzing spatial integration with reversing gratings[12,49,50], we used the signal amplitude of the second harmonic relative to the amplitude of the first. This was taken at the contrast combination that had the lowest first-harmonic amplitude, which indicates stimulation near the balance point. Using this measure, we found that chromatically linear and chromatically nonlinear cells as defined under spatially homogeneous stimulation had similar distributions of the grating nonlinearity index (Fig. 5f). In fact, for all distinguished subpopulations of cells, the large majority of grating nonlinearity indices were smaller than unity, indicating the absence of frequency doubling and thus mostly linear chromatic integration under this stimulus.

A potential reason for the discrepancy between the chromatic integration under spatially homogeneous and grating stimuli is that the spatially homogeneous stimulus strongly activates the surrounding of each ganglion cell's receptive field. For the grating, on the other hand, surround activation is likely weaker because, unlike for the receptive field center, there are multiple spatial grating periods over the surround, leading to at least partial cancellation. Thus, we hypothesized that nonlinear effects of chromatic integration as observed during spatially homogeneous stimulation arise from activation of the receptive field surround.

**Reducing surround activation with local stimuli linearizes chromatic integration.** To test the hypothesis that activation of the receptive field surround is critical for the occurrence of nonlinear chromatic integration, we implemented a spatially local version of the chromatic-integration stimulus, which aims at analyzing chromatic integration for a given cell with stimuli that are roughly restricted to its receptive field center. We, therefore, presented 500-ms flashes of local spots of 160 μm diameter with the same chromatic contrast combinations as for the full-field chromatic-integration stimulus (Fig. 2b). To cover the entire recording area, several randomly selected spot locations, separated by at least 320 μm, were used at the same time, each with a randomly selected contrast combination (Fig. 6a). For each recorded ganglion cell, we then constructed the color integration curves for each spot location and selected the location that induced the maximum response for further analysis (Fig. 6b). Note that stimulation at the optimal spot location essentially eliminates activation of the receptive field surround. At the same time, the spot may not fully cover the receptive field center, potentially reducing center excitation. Responses to the local stimulus may thus be higher or lower than responses to the full-field version, depending on the balance of reduced surround suppression and reduced center excitation. This does not affect our analysis, as chromatic integration in the center can also be investigated based on a partially activated receptive field center.

Sample responses of a linear cell and two nonlinear cells (On and Off, see Supplementary Fig. 4d–f for On–Off cells) indicate that chromatic integration was generally linear under local stimulation (Fig. 6c–e). Population analysis confirmed this observation. Comparing the nonlinearity indices of local and global stimulation showed that nonlinear effects of chromatic integration were strongly reduced for local stimuli

($p = 2.6 \times 10^{-23}$ for the population of all nonlinear cells; two-sided Wilcoxon signed-rank test; Fig. 6f), with only a few cells still crossing our threshold of ±0.1 for nonlinear chromatic integration.

The analysis above suggested that activation of the surround is critical to evoke nonlinear chromatic integration. To gain insight into which chromatic signals were carried by the receptive field surround at the balance point, we compared the relative sensitivity to UV light as compared to green light under local and full-field stimulation. This relative UV sensitivity can be obtained from the contrast values at the crossing point of the chromatic-integration curves (see "Methods" section) because the UV and green stimulus components were equally effective at this point. Figure 6g shows that going from full-field to local stimulation can considerably change the relative UV sensitivity. In particular, nonlinear Off and nonlinear On–Off cells generally were relatively more UV-sensitive under full-field stimulation. This indicates that the suppressive surround in these cells was relatively more sensitive to green as compared to the center, thus reducing overall green-sensitivity under full-field stimulation. Similar effects were also seen for nonlinear On cells, although the effect appears smaller and less clear, owing to the smaller number of recorded cells. Note, though, that nonlinear On–Off cells also contained a subset that was more green-sensitive under full-field stimulation; indicating that some of these cells may also bear a relatively more UV-sensitive suppressive surround.

**GABAergic and glycinergic inhibition are involved in nonlinear chromatic integration.** Given the observed role of the receptive field surround in generating nonlinear chromatic integration, we next probed the role of specific inhibitory interactions. To do so, we repeated the full-field chromatic-integration stimulus in the presence of different bath-applied inhibitory blockers. We found that gabazine, which blocks $GABA_A$ receptors, brought the chromatic nonlinearity indices of On, Off, and On–Off cells closer to zero and linearized chromatic integration in many of these cells (Fig. 7a, d), whereas no effect on the indices was found for linear cells. Application of TPMPA, which blocks $GABA_C$ receptors, did not affect the chromatic nonlinearity index of nonlinear Off cells (Fig. 7b, e), and reduced the indices for nonlinear On–Off cells, but did not make these cells completely linear (median nonlinearity indices: control = 0.25, drug = 0.18). Nonlinear On cells, on the other hand, nearly all became linear under TPMPA. Finally, blocking glycinergic inhibition with strychnine (Fig. 7c, f) had a strong linearizing effect on nonlinear On cells and also reduced nonlinearity indices of nonlinear On–Off cells significantly, whereas effects on nonlinear Off cells were small, though significant.

These data corroborate the importance of the inhibitory receptive field surround for generating nonlinear spectral integration. Blocking inhibitory interactions can strongly reduce or abolish the nonlinear effects. This suggests that amacrine cells are involved in the generation of nonlinear chromatic integration. Moreover, there seem to be cell-type-specific differences in the involvement of different inhibitory pathways. For nonlinear Off cells, inhibition through $GABA_A$ receptors appears most important, whereas chromatic integration in nonlinear On and nonlinear On–Off cells is also shaped by $GABA_C$ receptors and glycinergic inhibition.

**Chromatically nonlinear cells are predominantly localized in the ventral retina.** In the mouse retina, similar to other rodents, the expression of different cone opsins is not uniform but displays a spatial gradient across the retina. The ventral retina is

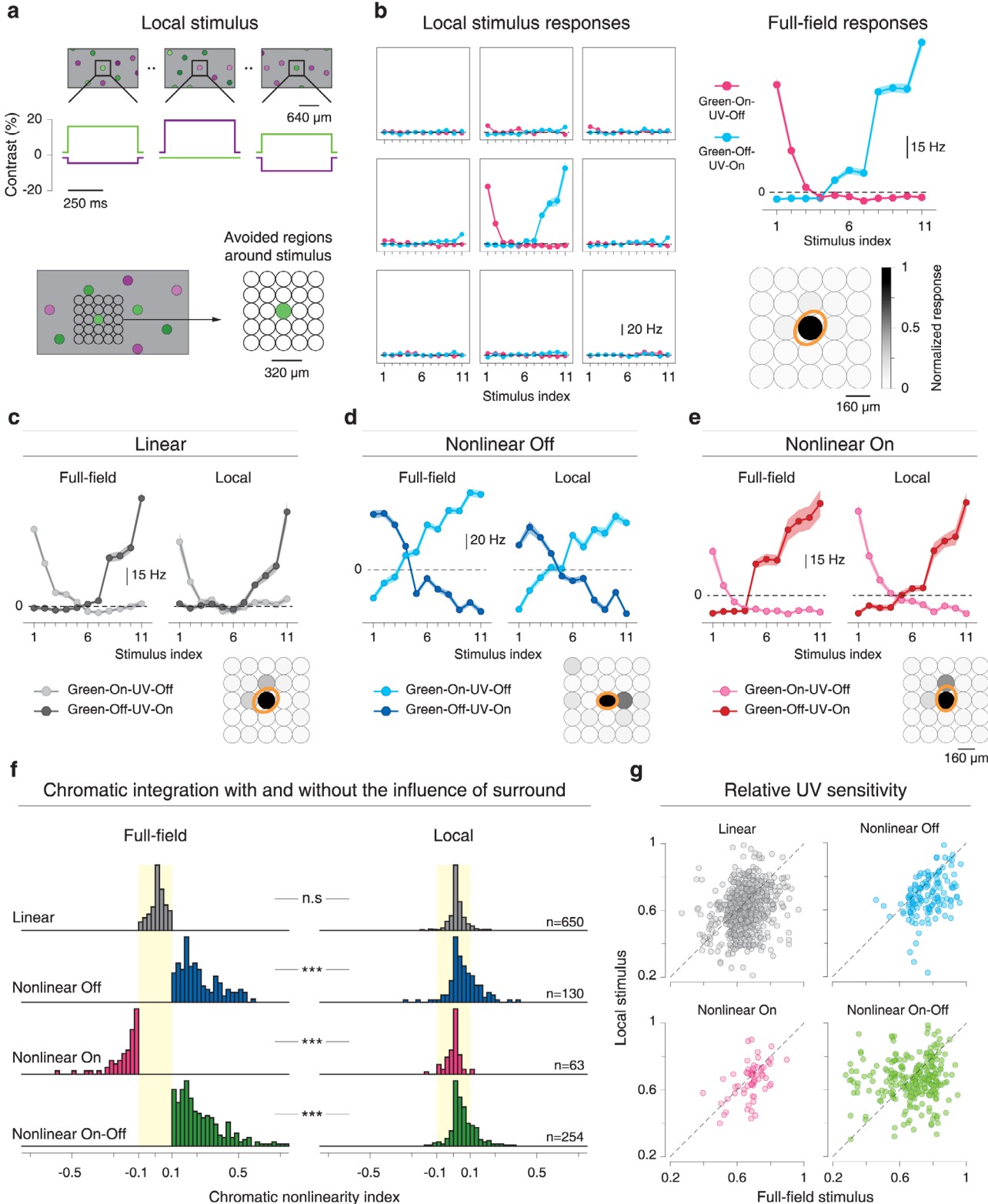

dominated by S-opsin and the dorsal retina by M-opsin[40,51]. Here, we aimed at investigating the relationship between the chromatic-integration properties of ganglion cells and their location across the dorsal–ventral axis of the mouse retina. To do so, we defined a retinal midline through the center of mass of all reconstructed receptive fields in our whole-mount recordings by using the occurrence of UV-selective cells (Supplementary

Fig. 2a) to identify the ventral side of the retina (see "Methods" section). We then measured the relative distance of each receptive field center to the midline, with negative values for cells on the ventral side. To check whether our approach to define the ventral retina was consistent with the opsin expression across the retina, we compared responses to UV and green light and related the relative chromatic preference of each ganglion cell, quantified by

**Fig. 6 Nonlinear chromatic integration is linearized under local stimulation. a** Schematic view of the local chromatic-integration stimulus, showing that multiple, spatially separated locations are stimulated simultaneously with different contrast combinations. Time courses of displayed contrast for one of the stimulation spots are shown below the frames of the top row. **b** Chromatic-integration curves from the responses of a sample cell for the receptive field center and eight stimulus grid locations around the center. Shaded regions around the curves show mean ± SEM. To the right, the cell's chromatic-integration curve for the full-field stimulus is shown for comparison (top), and the Gaussian fit to the receptive field (orange, 1.5-sigma contour) is displayed relative to the spot locations, with a grayscale-encoded maximum of the local chromatic-integration curve. **c–e** Chromatic-integration curves of a linear (**c**), a nonlinear Off (**d**), and a nonlinear On (**e**) cell, as determined from the full-field chromatic-integration stimulus. The display style is analogous to the example in **b**, but the local chromatic-integration curves are only shown for the location selected for analysis. Error bands show mean ± SEM. **f** Comparison of chromatic nonlinearity indices between full-field and local stimulation for linear, nonlinear Off, nonlinear On, and nonlinear On–Off cells. The distributions differ significantly for each of the nonlinear cell classes (linear cells: $p = 0.12$; nonlinear Off cells: $p = 8.9 \times 10^{-12}$; nonlinear On cells: $p = 5.1 \times 10^{-12}$; nonlinear On–Off cells: $p = 1.0 \times 10^{-41}$; two-sided Wilcoxon signed-rank test, statistics summary: ***$p < 0.001$, n.s. not significant). **g** Comparison of the relative sensitivity to UV stimuli under local versus full-field stimulation for the four distinguished cell classes.

a UV-green index[42], to the cell's position in the retina. For all the retinas included in this analysis, the defined dorsal–ventral axis of the retinas was consistent with the gradient of UV-green index values, displaying stronger UV sensitivity in the identified ventral retina.

Figure 8a shows a sample retina together with the layout of recorded receptive fields and chromatic-integration curves for some of the cells. The data demonstrate that chromatically linear and nonlinear cells were not equally distributed across the retina. Linear cells were predominantly found in the dorsal part of the retina, whereas nonlinear cells often occurred in the ventral retina (Fig. 8b and Supplementary Fig. 5). This observation is corroborated by pooling the data over all analyzed retinas ($N = 17$) and comparing the distributions of linear and nonlinear cells along the dorsal–ventral axis (Fig. 8c, see also Supplementary Fig. 5c); the distribution for nonlinear cells is skewed towards the ventral direction, and the opposite is true for the distribution of linear cells.

**Nonlinearity of chromatic integration is driven by the rod pathway.** The prevalence of nonlinear chromatic integration in the ventral retina raises the question about the origin of the green-sensitive signals that contribute to the chromatic integration. As the green-sensitive M-opsin occurs only sparsely in the ventral retina, we hypothesized that it is the rod photoreceptors that provide green-sensitive signals. Unlike the M-opsin, rods are distributed evenly along the dorsal–ventral axis of the retina. Moreover, previous studies showed that the rod pathway in the mouse retina can contribute to chromatic signaling even under photopic conditions by providing green-sensitive signals to color-opponent processing in some ganglion cells[52,53].

To investigate the role of rod photoreceptors in nonlinear chromatic integration, we tested the effect of different light levels by comparing responses to the full-field chromatic-integration stimulus at our standard light level in the mesopic/low photopic regime and at a 10-fold increased intensity (mid-to-high photopic). These experiments showed that higher light levels tended to linearize the chromatically nonlinear cells. This effect was similar for all classes of nonlinear cells (Fig. 9a, b), while linear cells generally remained linear under higher light levels. This is consistent with our hypothesis that the rod pathway is critical for nonlinear chromatic integration.

Recent studies have shown that one relevant pathway for rod signals under photopic conditions is the rods' connection to cone photoreceptors via horizontal cells[54]. This pathway has been shown to provide the green-sensitive surround of some color-opponent ganglion cells in the mouse retina[52,53]. To inspect the role of horizontal cells in nonlinear chromatic integration, we applied the pH buffer HEPES to block the feedback from horizontal cells to photoreceptors[52,54,55]. We observed that nonlinear Off and On–Off cells became more linear under the

application of HEPES (Fig. 9c, d), whereas the nonlinearity of On cells was not affected by blocking horizontal cells. Thus, horizontal cells may contribute to nonlinear chromatic integration for some ganglion cells, specifically Off and On–Off cells, but do not provide a general mechanism of chromatic nonlinearity, as chromatic nonlinearities in On cells appear independent of horizontal cells. These data also further underscore the difference between nonlinear On cells on the one hand and nonlinear Off and On–Off cells on the other hand.

**Functional consequences of nonlinear chromatic integration.** The measured nonlinear chromatic integration should primarily affect responses to stimuli for which two chromatic signals have opposing contrast. Although this condition is easily met in the experimental setup, the occurrence in natural scenes is unclear. To test what the effects may be for the encoding of natural scenes, we built simple models of linear and nonlinear chromatic integration and passed colored images of natural scenes through each model (Fig. 10a). For concreteness, we simulated nonlinear chromatic integration by filtering the green and UV components of the images separately with Off-type Gaussian receptive fields and combining the two resulting signals after half-wave rectification. This phenomenological model captures the elicited activity under stimuli with opposing contrast in green and UV illumination in a generic fashion without the need to specify a particular circuit mechanism. For comparison, we simulated chromatically linear cells with the same structure, but without the rectification of the two chromatic signals.

As input, we took images from a database[56,57] of natural scenes with separate UV and green color channels (Fig. 10b), and converted the pixel intensities to Weber contrast for each color channel. For each image, we randomly selected 10,000 patches to examine where the response of the nonlinear model differed from that of the linear model (Fig. 10c, e). For many patches, the two models yielded identical responses, as expected when green and UV contrast had the same sign. For some patches, however, in particular, along with the separation of sky and foreground, the nonlinear model displayed a distinctly stronger response.

In order to understand how the skyline is marked by nonlinear chromatic integration, we inspected the contrast signals in the UV and green channels along vertical lines in the image (Fig. 10d). When transitioning from the bright sky down to the dimmer foreground, both contrast signals drop from positive to negative values, but the UV signal does so earlier (Fig. 10f), likely because of stronger reflectance of green light from the foliage whereas UV light is effectively blocked. This leads to a transition zone with opposing contrast of UV and green, and the resulting higher activity of chromatically nonlinear cells as compared to linear cells in this region could help detect the skyline for spatial orientation.

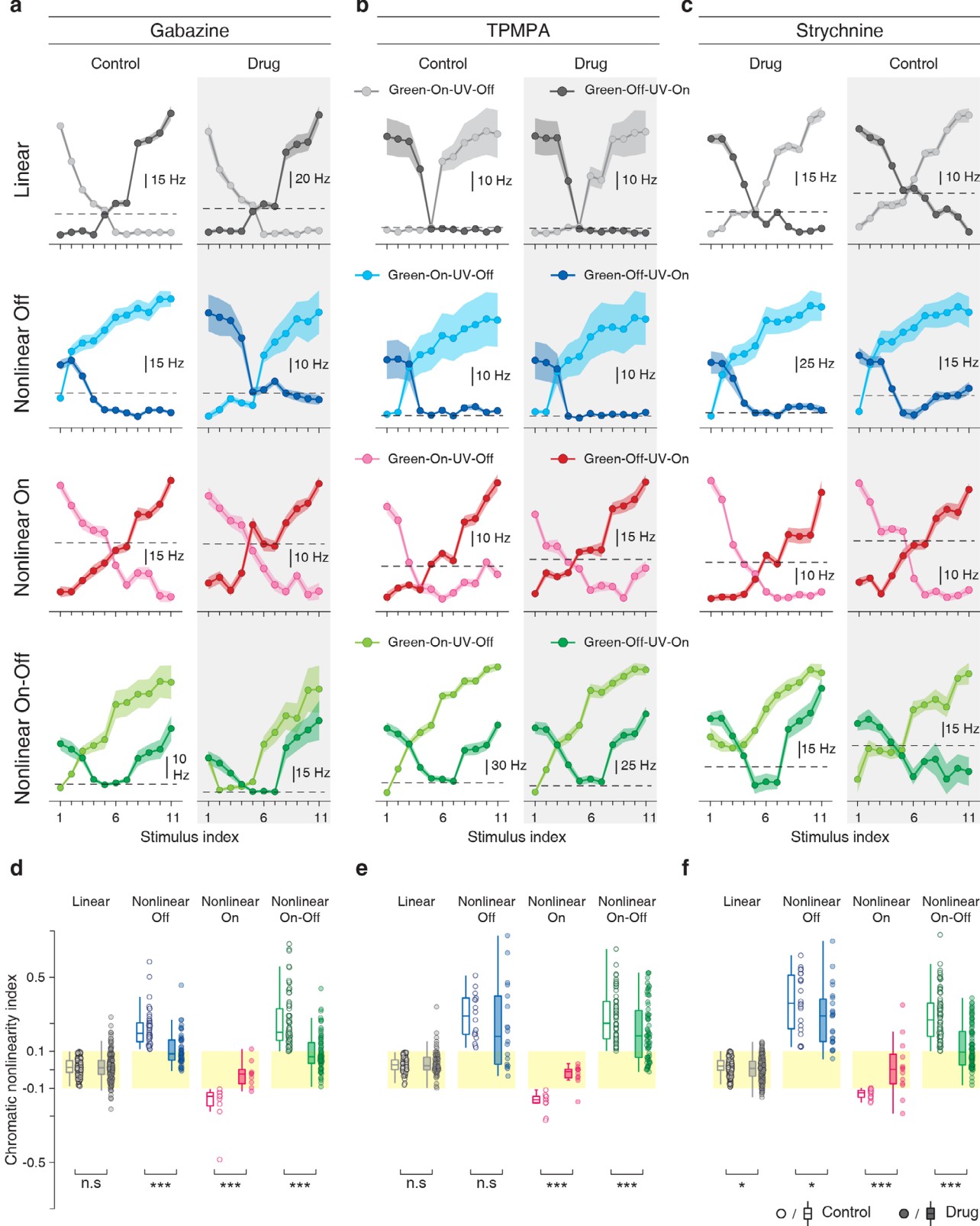

## Discussion

Nonlinear signal integration across time and space plays an essential role in different computational functions of the retina[9,58,59]. In addition, retinal ganglion cells also integrate signals along the chromatic dimension. To study chromatic integration, we designed a novel stimulus, which presents a series of opposing contrasts of UV and green colors, and searched for a balance point in the responses to one contrast combination and its reversed version. We found chromatically linear cells that did not respond at the balance point and nonlinear cells that responded strongly to the stimuli presented at the balance point (Figs. 1 and 2). A striking pattern among

**Fig. 7 GABAergic and glycinergic inhibitory interactions shape the nonlinearities of chromatic integration. a** Chromatic-integration curves of sample linear, nonlinear Off, nonlinear On, and nonlinear On–Off cells for full-field stimulation before and during application of a GABA$_A$ blocker (gabazine). **b** Same as **a**, but for the GABA$_C$ blocker TPMPA. **c** Same as in **a** and **b**, but for the glycine blocker strychnine. Shaded regions around the curves show mean ± SEM. **d** Distributions of chromatic nonlinearity indices before and during application of gabazine. Indices of linear cells remained unchanged but were strongly reduced for all nonlinear cell classes (linear cells: $p = 0.64$, $n = 194$: nonlinear Off cells: $p = 1.3 \times 10^{-8}$, $n = 47$; nonlinear On cells: $p = 3.9 \times 10^{-4}$, $n = 9$; nonlinear On–Off cells: $p = 3.4 \times 10^{-12}$, $n = 71$). **e** Same as **d**, but for TPMPA. Both nonlinear On and On–Off cells displayed a significant effect (linear cells: $p = 0.81$, $n = 87$; nonlinear Off cells: $p = 0.06$, $n = 18$; nonlinear On cells: $p = 1.9 \times 10^{-3}$, $n = 10$; nonlinear On–Off cells: $p = 1.8 \times 10^{-6}$, $n = 71$). **f** Same as **d** and **e**, but for strychnine. This strongly affected nonlinear On and nonlinear On–Off cells, and had weaker effects on nonlinear Off and linear cells (linear cells, $p = 0.02$, $n = 180$, nonlinear Off cells, $p = 0.02$, $n = 23$, nonlinear On cells, $p = 3.4 \times 10^{-4}$, $n = 13$, nonlinear On–Off cells, $p = 4.4 \times 10^{-13}$, $n = 89$). All statistical comparisons are based on the two-sided Wilcoxon signed-rank test (statistics summary: ***$p < 0.001$, *$p < 0.05$, n.s. not significant). For all boxplots, the central line and the box mark the median and the interquartile range (IQR) from the first to the third quartile, respectively, and whiskers extend to the maximum and minimum within the central range of 1.5 × IQR. The original data points are displayed next to each boxplot.

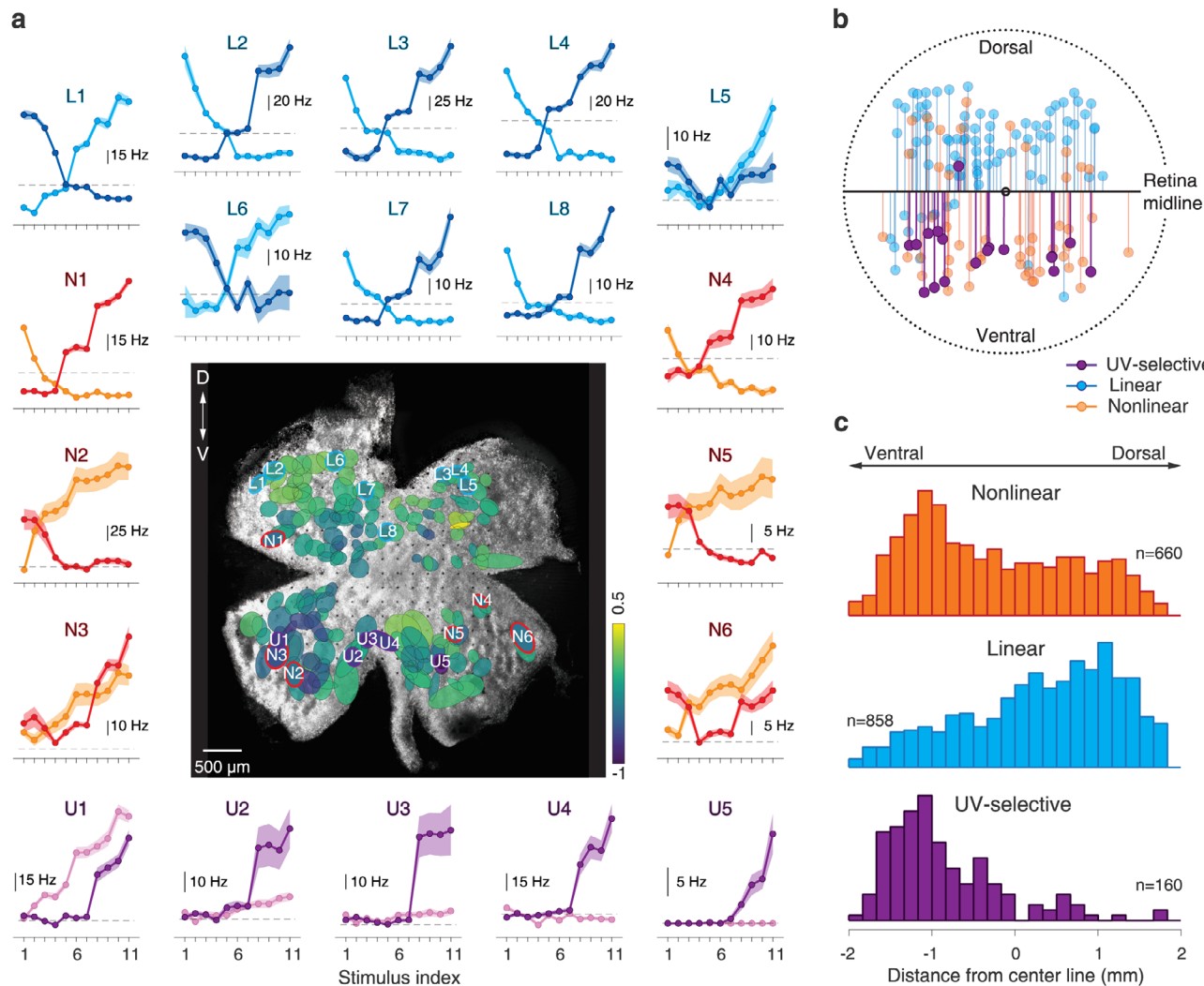

**Fig. 8 Locations and potential functions of chromatically nonlinear cells. a** Image of a sample retina with ganglion cell receptive fields (ellipses corresponding to the 1.5-$\sigma$ contour of a fitted Gaussian with color indicating the UV-green index) and chromatic-integration curves for some of the cells. The labels identify the cells as linear (L...), nonlinear (N...), or UV-selective (U...) and mark the position of the receptive field on the retina. Shaded regions around the curves show mean ± SEM. **b** Distances of receptive field centers to the midline for the retina shown in **a**. **c** Distribution of receptive field distances from the midline for chromatically nonlinear, linear, and UV-selective cells over all analyzed retinas. Note that the prevalence of UV-selective cells in the ventral retina is by methodological design, as these cells were used to identify the ventral side, and the corresponding distribution is shown here only for comparison.

nonlinear cells was that Off and On–Off cells showed positive responses at the balance point, whereas On cells displayed response suppression (Fig. 3). Furthermore, nonlinear chromatic integration occurred independently of nonlinear spatial integration (Fig. 4) and depended on stimulation of the

inhibitory receptive field surround, as gratings (Fig. 5), local stimuli (Fig. 6), and certain inhibition blockers (Fig. 7) linearized responses. Finally, we showed that chromatically nonlinear cells are predominantly located in the ventral retina (Fig. 8), that the nonlinearity depends on green-sensitive inputs from

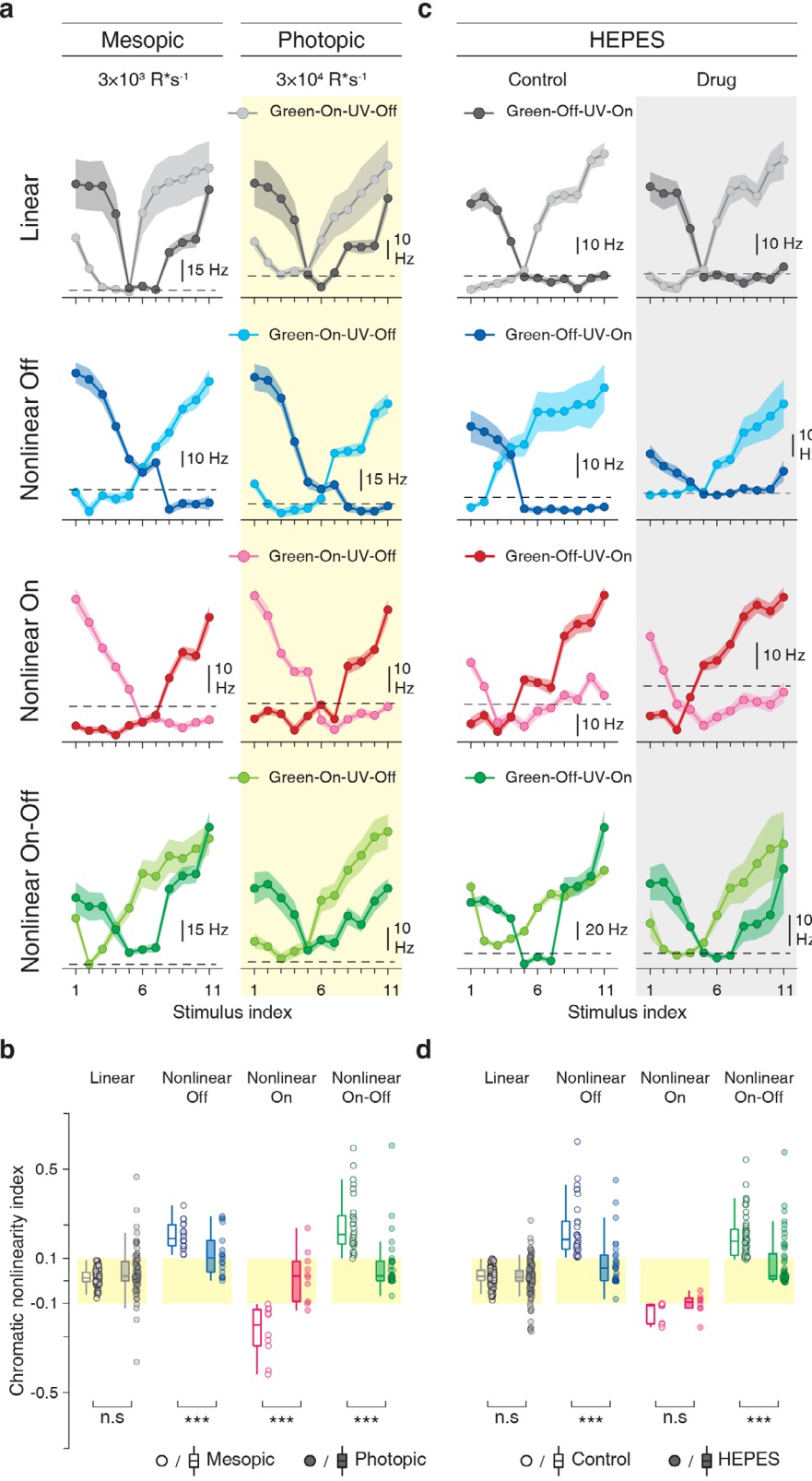

the rod pathway (Fig. 9), and that these cells may aid detection of the skyline where green and UV light occurs with opposing contrast (Fig. 10).

A central observation was the difference in the activity of nonlinear On versus Off and On–Off cells at the balance point. The increased firing rates in nonlinear Off and On–Off cells are reminiscent of the characteristic responses of spatially nonlinear Y cells under reversing gratings, but the suppressed responses of nonlinear On cells represent a novel signature. The consistency with which these patterns were observed (all of 144 nonlinear On cells showed response suppression at the balance point and all of 328 nonlinear Off and 567 On–Off cells showed increased

**Fig. 9 Rod pathway is involved in the nonlinearities of chromatic integration. a** Chromatic-integration curves of sample linear, nonlinear Off, nonlinear On, and nonlinear On–Off cells for full-field stimulation during mesopic and photopic light levels. **b** Distributions of chromatic nonlinearity indices during mesopic and photopic brightness levels. Indices of linear cells remained unchanged but were reduced for all nonlinear cell classes (linear cells: $p = 0.02$, $n = 75$; nonlinear Off cells: $p = 4.9 \times 10^{-3}$, $n = 18$; nonlinear On cells: $p = 3.9 \times 10^{-3}$, $n = 10$; nonlinear On–Off cells: $p = 1.1 \times 10^{-4}$, $n = 29$). **c** Same as **a**, but (instead of the increased light level) for application of the pH buffer HEPES, which blocks the influence of the horizontal cells in the retina. Shaded regions around the curves show mean ± SEM. **d** Same as **b**, but before and during application of HEPES at the same (mesopic) light level. Nonlinear Off and On–Off cells displayed a significant effect (nonlinear Off cells: $p = 1.2 \times 10^{-3}$, $n = 31$; nonlinear On–Off cells: $p = 1.7 \times 10^{-6}$, $n = 50$) while nonlinear On and linear cells did not change their chromatic-integration properties (nonlinear On cells: $p = 0.15$, $n = 8$; linear cells: $p = 0.21$, $n = 156$). All statistical comparisons are based on the two-sided Wilcoxon signed-rank test (statistics summary: ***$p < 0.001$, *$p < 0.05$, n.s. not significant). For all boxplots, the central line and the box mark the median and the interquartile range (IQR) from the first to the third quartile, respectively, and whiskers extend to the maximum and minimum within the central range of 1.5 × IQR. The original data points are displayed next to each boxplot.

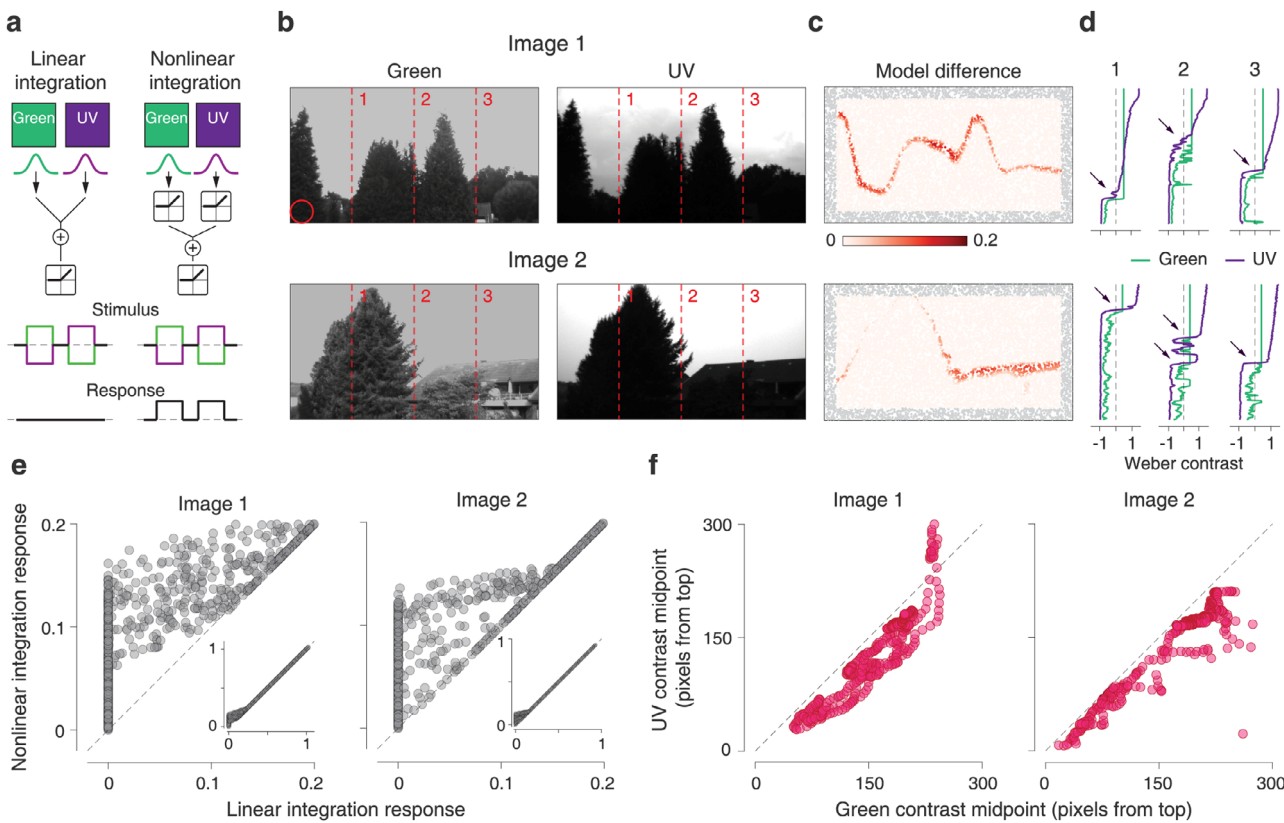

**Fig. 10 Potential functions of chromatically nonlinear cells. a** Schematic depiction of linear and nonlinear models of chromatic integration. **b** Examples of green (left) and UV (right) channels in natural images. The red circle shows the receptive field size used in both models. **c** Heatmap of response differences from the two models (nonlinear model minus linear model) for 10,000 randomly selected patches of the images in **b**. Gray points show locations where the modeled receptive field was partly outside the image range. **d** Contrast values for the UV and green channels along the three columns marked by the dashed lines in **b**. Arrows point towards regions of opposite contrast in the two color channels. **e** Comparison of the two model responses for low to moderate activation. The inset shows the same measurement over the entire activation range. **f** Comparison of midpoints for fitted sigmoids to the contrast values of the green and UV channels along with all columns in the images of **b**.

activity) suggests that this is a highly specific property, presumably because the chromatically nonlinear cells comprise specific subtypes of ganglion cells.

Despite the hypothesis that nonlinear chromatic integration is cell-type specific, the measured chromatic nonlinearity indices did not fall into discrete groups but formed a continuum, reminiscent of characterizations of spatial integration[50,60]. Experimental variability likely is a factor here, but other contributions could come from the variability of nonlinear effects within cell types, for example, variability that is linked to a cell's location along the dorsal–ventral axis of the retina. In addition, the limited range of contrast values tested here may not have effectively activated the surround of all cells. A challenge for a more detailed

cell-type-specific analysis is the difficulty to clearly identify cell types in extracellular multielectrode-array recordings. Let us note, however, that interesting structure can already be seen for two readily identifiable functional classes of ganglion cells, direction-selective (DS) and orientation-selective (OS) cells (Supplementary Fig. 6). DS cells were nearly always chromatically linear, whereas OS cells appeared to fall into two groups, one chromatically linear and the other nonlinear. Interestingly a similar division into linear and nonlinear OS cells was also observed with respect to spatial integration[61] and might relate to different types of OS cells[62]. Clearly, more work will be required to better relate these different functional properties of OS cells and establish the cell-type specificity of nonlinear chromatic integration.

Unlike the frequently observed nonlinear spatial integration in mouse retinal ganglion cells[50,60], nonlinear chromatic integration only occurred in a minority of our recorded cells, and even these became linear when stimulation was restricted to the receptive field center. A potential explanation of the abundance of linear chromatic integration could be that the integration starts already at the level of photoreceptors. Many photoreceptors in the mouse retina co-express S- and M-opsins[51], and the two opsin-mediated signals should here linearly combine. In addition, cone photoreceptor signals are thought to be linearly integrated by postsynaptic bipolar cells[15], and the majority of the bipolar cells in the mouse retina receive mixed inputs from the different cone types[23,25], providing another level of linearization of chromatic integration. Note, however, that there are also exceptions within the mouse retinal circuitry. Some fraction of cones exclusively expresses S-opsins, and the so-called type-9 bipolar cell, an On-type cell, collects signals exclusively from these pure S-cones[63]. Conversely, type-1 bipolar cells, which are Off-type, primarily sample M-cone input[23,25], making them particularly sensitive to green or even green-selective, at least in the dorsal retina. However, we found that nonlinear chromatic integration is particularly pronounced in the ventral retina, where M-opsins are scarce, and that it depends on rod signals. This suggests that rod-bipolar cells might be the critical source of green-sensitive (On-type) signals. Alternatively, direct interactions of rods with Off-type cone bipolar cells[64,65] and rod-cone interactions via horizontal cells[54] might also contribute, whereas rod-cone gap junctions[66,67] likely only yield linear chromatic integration. Future experiments at low mesopic light levels, where rod signals should be relatively stronger[68], may help further elucidate how the rod pathways contribute to nonlinear chromatic integration.

How exactly the color-sensitive signals are nonlinearly combined by some ganglion cells is an intriguing question about which we can only speculate at the moment. Yet, the nonlinearity appears to rely on inhibitory amacrine cells in the receptive field surround, as indicated by the linearization under local stimulation and pharmacological blockers of inhibition. The crucial role of inhibition also fits well with the observation of activity suppression at the balance point in chromatically nonlinear On cells. This might be produced, for example, by a chromatically nonlinear amacrine cell, which provides inhibition at the balance point while linear center excitation cancels out. Alternatively, the amacrine cell might also be chromatically linear, yet interact nonlinearly with the ganglion cell, which could lead to extra inhibition near the balance point if the amacrine cell has a different relative UV/green sensitivity (i.e., has itself a different balance point) than the receptive field center, potentially because it receives relatively more rod-mediated signals. The latter scenario is in line with our observation that most nonlinear cells appear to have a suppressive surround that is more green-sensitive than the center (Fig. 6g). Thus, for the chromatically nonlinear On cells, nonlinear and green-sensitive surround suppression could come from amacrine cells that receive green-sensitive On-type signals from rod-bipolar cells (Fig. 9), similar to other suggested circuits of color processing[52,53]. Note that this is also consistent with the robustness of nonlinear chromatic integration in On cells under the suppression of horizontal feedback.

Compared to the response suppression of chromatically nonlinear On cells, the increased activity of nonlinear Off (and On–Off) cells at the balance point appears more difficult to explain through a mechanism that is based on the inhibitory surround. If center stimulation alone leads to no evoked activity at the balance point, adding inhibitory input from the surround should not lead to increased activity. This holds even if the balance point shifts when going from local to global stimulation, as on each side of the balance point, one of the two local chromatic-integration curves should be at or below zero activity. Thus, the surround should contribute an excitatory component, which then takes part in generating the nonlinear chromatic integration. Although there are many circuit motifs that could provide such a signal (e.g., lateral excitation through glutamatergic amacrine cells[69]), a parsimonious explanation can be given by a serial-inhibition circuit that involves the same inhibitory green-sensitive On-type signal that we hypothesized for the chromatically nonlinear On cells. If this On-type inhibition, for example, acts on amacrine cells in the receptive field surround of a ganglion cell, the surround's sensitivity will be biased towards a green Off signal, as required to explain the ganglion cell's sensitivity shift towards UV under global stimulation (Fig. 6g). And nonlinear signal transmission along this serial inhibition to the ganglion cell would create nonlinear chromatic integration by the ganglion cell. However, the dependence of nonlinear chromatic integration in Off cells on horizontal cell signaling suggests that the mechanism may involve further interactions in the surround or simply break down when the balance of excitatory and inhibitory signals in the surround is disturbed.

Functionally, comparison of relative intensities of UV and green light may provide an important navigational signal and is used by insects such as ants to detect salient features of the environment like the skyline for visual navigation[70–73]. Our model analysis of responses to natural images showed that chromatically nonlinear cells may contribute to this skyline detection in the mouse. Fittingly, these cells are predominantly located in the ventral retina, where they receive visual signals from the skyline above the vegetation, which may aid the control of gaze direction for alignment with the horizontal plane[74] and for continuous monitoring of the overhead space[75].

Finally, many standard models of visual responses in retinal ganglion cells assume linear receptive fields and thus linear integration of visual signals. Yet, recent studies have emphasized the importance of incorporating nonlinear signal integration, in particular nonlinear spatial integration, to predict the responses of ganglion cells to natural stimuli[18,19,76–78]. Commonly, such models are fitted to responses from achromatic (grayscale) stimuli, and it will be interesting to explore how to include nonlinear chromatic integration for predicting response to chromatic stimuli. As part of this endeavor, a challenge will be to investigate whether it is possible to identify subunits of nonlinear chromatic integration. This would require identifying model circuit elements that linearly integrate over the spectral components of the stimulus with some spectral sensitivity and finding an appropriate nonlinear interaction of these chromatic subunits. However, given that nonlinear chromatic integration relies on inhibition, the subunit interactions likely need to be more complex than the rectified summation in standard models of nonlinear spatial integration.

## Methods

**Tissue preparation and electrophysiology**. We used retinas of adult wild-type mice (C57BL/6; aged 8–13 weeks) of either sex obtained from Charles River Laboratories Germany and housed at 20–24 °C with 50–70% humidity on a 12-h light/dark cycle. All the experiments and procedures conformed to national and institutional guidelines and were approved by the institutional animal care committee of the University Medical Center Göttingen (protocol number T11/35). Before experiments, mice were dark-adapted for at least 1 hour, and after dark adaptation, they were euthanized by cervical dislocation. The eyes were removed quickly and transferred to a chamber with oxygenated (95% $O_2$ and 5% $CO_2$) Ames' medium (Sigma-Aldrich, Munich, Germany), buffered with 22 mM $NaHCO_3$ (to maintain pH of 7.4) and supplemented with 6 mM D-glucose. The eyes were dissected and the cornea, lens, and vitreous humor were carefully removed for direct access to the retina. The retina was then isolated from the pigment epithelium and transferred to a multielectrode array (MultiChannel Systems, Reutlingen, Germany; 252-electrode planar or 60-electrode perforated arrays, 30 μm electrode diameter, and 100 or 200 μm minimum electrode spacing). During the recording, the retina was constantly perfused with the oxygenated Ames'

medium (4–5 ml/min), and the temperature of the recording chamber was kept constant around 32–34 °C, using an inline heater (PH01, MultiChannel Systems, Reutlingen, Germany) and a heating element below the array (controlled by TCX-Control 1.3.4, MultiChannel Systems). In some experiments, the retina was dissected into two halves, and only one half of the retina was mounted on the multielectrode array. The remaining retina pieces were stored in a chamber perfused with oxygenated Ames' medium for later recordings. All the experiments and tissue preparations were done in a dark room. The retina preparations were performed under infrared illumination with a stereomicroscope equipped with night-vision goggles.

The signals from the ganglion cells were amplified, band-pass filtered (300 Hz to 5 kHz), and stored digitally at 25 kHz (60-electrode arrays) or 10 kHz (252-electrode arrays) using MC-Rack 4.6.2 software (MultiChannel Systems). A custom-made spike sorting program based on a Gaussian mixture model and an expectation-maximization algorithm[79] was used to extract the spikes from the recorded voltage traces. Only units with a clear refractory period and a well-separated cluster of voltage traces were included in the final analysis. In total, 3346 ganglion cells from 31 retinas of 19 mice were used in the analysis.

**Dorsal–ventral orientation of the mouse retina**. We determined the dorsal–ventral axis by using the location of the UV-selective cells, which are thought to occur mainly in the ventral retina. For this analysis, we only included whole-mount retinas with recovered receptive fields that ranged across the entire retina and with at least 3 UV-selective cells (17 out of 31). We first defined the center of the retina by calculating the center of mass for all recorded receptive fields. We then determined the midline that passed through the center so that most UV-selective cells lay on one side (ventral side) with maximal distance from the line. Concretely, the angle of the midline was chosen so that the ratio between the mean signed distance of receptive field centers to the line and the standard deviation of the distances was maximal. For further analysis, we measured the distance of each receptive field center to the midline in order to compare the distribution of distances for chromatically linear and nonlinear cells.

We also checked the orientation of the retina by identifying landmarks in the choroid of the mouse eye[80] during retina preparation and, in some experiments, by making a burn-mark on the outer surface of the eye before enucleation and using this to mark the dorsal–ventral axis of the retina. These alternative procedures to determine retina orientation roughly agreed with our method of using UV-selective cells but were not always unambiguously identifiable (2 retinas) under the illumination conditions during our preparation procedure.

For visualization of receptive fields with respect to the location on the retina (Fig. 8a and Supplementary Fig. 5a), a picture of the recorded retina was taken at the end of each experiment with a microscope camera (image taken with either ThorCam 3.5.1 (Thorlabs) or Pylon5 5.0.0 (Basler AG) software). The receptive field centers were aligned to the retina pictures with a Procrustes analysis (using the Matlab procedure "procrustes"), which mapped receptive field outlines (given in coordinates of the stimulation screen, as obtained from the spike-triggered average) onto the pixel coordinates of the retina image. To do so, the position coordinates of the electrodes in the image were identified by marking the outline of the recording area with a rectangular region of interest on the image, and each recorded ganglion cell was associated with the electrode position that carried the primary signal of the cell. Receptive field center points (in units of pixels of the stimulus screen) and corresponding electrode positions (in units of image pixels in the region of interest) were gathered in the rows of two matrices $X$ and $Y$, respectively, whose two columns contained the two spatial coordinates. The Procrustes analysis then determined the transformation $Z = bXT + c$ that maps the receptive field coordinates $X$ into corresponding image coordinates, where $b$ is a scaling factor, $T$ is an orthogonal rotation matrix, and $c$ is a matrix with constant entries in each column to shift all receptive fields together. These parameters were optimized by minimizing the least-squares difference between transformed receptive field centers $Z$ and corresponding electrode positions $Y$, and the obtained transformation was used to plot receptive field outlines onto the images.

**Pharmacology**. All applied pharmacological drugs were prepared freshly from stock solutions for each experiment. The drugs were diluted in oxygenated Ames' medium (see above) and bath-applied to the retina. Recordings were resumed after 10–15 min. The following concentration of the drugs were used: either 5 or 10 μM GABAzine (SR95531; Sigma-Aldrich), 50 μM TPMPA ((1,2,5,6-tetrahydro-pyridine-4-yl)methylphosphinic acid; Sigma-Aldrich), either 0.5 or 1 μM of strychnine (Sigma-Aldrich), and 20 mM HEPES (4-(2-hydroxyethyl)-1-piperazineethanesulfonic acid; Sigma-Aldrich). Before administration of HEPES to the retina, the pH of the Ames' solution was adjusted to 7.4.

**Light stimulation**. Visual stimuli were generated and controlled through custom-made software, written in C++ and using the OpenGL library. Dichromatic light stimuli were displayed with custom-built UV-green projectors. The projectors were modified versions of DLP lightcrafter projectors (evaluation module (EVM), 864 × 480 pixels, Texas Instruments Company, USA), where the blue LED was replaced by a UV LED (peak, 365 nm, Nichia NCSU276A, Nichia, Japan). The stimuli were gamma-corrected, de-magnified (8 μm pixel size on the retina), and displayed by the projector at 60 Hz onto the photoreceptor layer of the retina.

For calibration of the display, the intensity of the projector was measured using a photodiode amplifier (PDA-750 Photodiode Transimpedance Amplifier, Tetrahertz Technologies Inc., USA), and the spectrum of each LED was measured with a spectrometer (CCS200 Thorlabs, USA). Photoreceptor isomerization rates were calculated using the relative sensitivity of the mouse opsins obtained from the known opsin sensitivity profiles[37,39], the measured LED spectra (Supplementary Fig. 1b), and a collecting area of 0.2 μm for cones[38] and 0.5 μm for rod photoreceptors[81]. Experiments were performed using two projector systems with slightly different brightness values. At mean brightness, the irradiance from the UV LEDs of the two setups used in the experiments was 5.06 and 5.2 mW/m², respectively. This led to ~1050 and ~1100 isomerizations per S-cone per second and ~420 and ~450 co-isomerizations per M-cone per second. Irradiance of the green LEDs at mean brightness was 1.68 and 1.89 mW/m², causing ~680 and ~660 isomerizations per M-cone per second and negligible (fewer than 5) co-isomerizations per S-cone per second. The overall rod isomerizations were ~3000 per rod per second at mean intensity level, using both light sources together. For the experiments at the higher brightness level (Fig. 9a, b), we increased the brightness of both LEDs by 10-fold after removing a neutral density filter (optical density = 1.0) from the light path. This increased the rod isomerizations to ~30,000 per rod per second at the mean intensity level. All applied monochromatic stimuli, such as reversing gratings and spatiotemporal white noise, used both LEDs with the same mean brightness values as stated above and jointly modulated over space and time with identical contrast values.

**Opsin-isolating stimuli**. In order to compensate for the co-activation of S- and M-opsins by green and UV light, respectively, we used the method of silent substitution[35] to generate S- and M-opsin-isolating stimuli. In this method, for every contrast of UV light, an opposite contrast of green light is shown to compensate for the activation of M-opsins induced by the UV light. Given the similar spectral sensitivities of M-opsins and rod-opsins[36–38], this will also diminish the activation of rods when using an S-opsin-isolating stimulus, whereas M-opsin-isolating stimuli will co-activate rods with approximately the same contrast. To obtain stimuli with opsin-isolating contrast, we first determined the transfer matrix $A$ between illumination intensities and isomerization rates:

$$A = \begin{bmatrix} S_{uv} & S_{green} \\ M_{uv} & M_{green} \end{bmatrix},$$

where $S_{uv}$ and $M_{uv}$ are the isomerization rates of S- and M-opsins under illumination with UV light and $S_{green}$ and $M_{green}$ are the isomerization rates under green light. We then used the inverted matrix $A^{-1}$ to determine the required changes in UV and green illumination to obtain the desired change in isomerization rates of S- and M-cones. All the dichromatic stimuli used in this work were opsin-isolating, and we speak of "UV contrast" and "green contrast" as a shorthand to refer to relative changes in activation of S-opsins on the one hand and M-opsins as well as rod-opsins on the other hand.

**Receptive field measurements**. The receptive field of each ganglion cell was measured by calculating the spike-triggered average in response to a monochromatic spatiotemporal binary white-noise stimulus (either 75 or 100% contrast and either 30 or 60 Hz update frequency), which was displayed on a checkerboard layout with squares of either 48 or 60 μm to the side. The spike-triggered average was decomposed into its temporal and spatial components using singular value decomposition[82,83]. We fitted the spatial component of the receptive field by a two-dimensional Gaussian function, which we used to identify the center point of the receptive field and to define the receptive field diameter as the diameter of a circle with the equal area as inside of the 1.5-$\sigma$ contour of the Gaussian fit.

**Chromatic-integration stimulus**. To assess the chromatic-integration properties of mouse ganglion cells, we used a UV-green full-field stimulus. This stimulus consisted of 22 different contrast combinations of UV and green light, which were presented in a step-like fashion for 500 ms each, separated by either 1.5 or 2 s of background illumination. Each contrast combination was presented on average 50 times, and the order was determined by a Fisher–Yates random permutation algorithm[84] to ensure randomized, unbiased sampling of all contrast combinations. The contrast combinations were categorized into two sets of green-On-UV-Off and green-Off-UV-On, respectively (Fig. 2b), and contrast values refer to Weber contrast $C = (I - I_b)/I_b$, where $I$ is the applied stimulus intensity and $I_b$ is the background intensity. The green-On-UV-Off set consisted of 11 different combinations, ranging from 20% green and 0% UV to 0% green and −20% UV in steps of −2% for both colors. The green-Off-UV-On set contained the contrast-reversed combinations. The design of the stimulus aimed at being analogous to shifting a black/white edge across the receptive field, as used to study spatial integration.

To analyze the responses of a recorded ganglion cell, we computed peri-stimulus time histograms (PSTHs) with 10 ms bin size for all 22 contrast combinations. To construct the chromatic-integration curves, we calculated the average firing rate $R$ from 50 to 250 ms after onset of each contrast combination.

We did not use the entire stimulus duration because many cells showed transient responses with activity suppressed during the second half of the stimulus presentation. We subtracted the baseline firing rate, which was measured during the 200 ms that preceded the contrast combination. All further analyses of responses to the chromatic-integration stimulus were based on this baseline-subtracted response measure. We plotted these firing rates so that every contrast combination was aligned on the x axis with its contrast-reversed combination. We then looked at the crossing point of the two curves, which identified a balance point where the response to one contrast combination and its reversal were the same.

**Chromatic nonlinearity index.** To quantify the degree of chromatic nonlinearity for each cell, we defined a chromatic nonlinearity index. The index aimed at measuring the response at the balance point where a green-On-UV-Off stimulus and its contrast-reversed version yielded the same response. For this purpose, we searched for a crossing of the green-On-UV-Off and green-Off-UV-On response curves. A crossing point was identified by two neighboring locations in these plots where one curve had a higher value than the other curve at one location and a lower (or equal) value at the other location. For cells with more than one crossing point, the index was calculated from the crossing point closest to zero. Cells without a crossing point were characterized as either UV-selective or color-opponent (see below), with no chromatic nonlinearity index.

In order to be conservative in our measure of activity at the crossing point, we used a lower bound on the activity, obtained by only assuming that the two curves run monotonically between the data points that delimit the crossing point. Concretely, we first estimated for each of the two curves the minimum activity level (disregarding the sign of the activity) in the interval between the two data points around the crossing point. From the two activity levels at the two data points, this is the one that is closer to zero, unless the activity levels at the two data points have opposite sign, in which case it is zero. Out of the two selected activity values from the two curves, the lower bound on the crossing point was given by the one that was most distant from zero. Finally, we normalized this lower-bound measure of the activity at the crossing point by the maximum response over all 22 stimuli in the chromatic-integration stimulus sequence in order to obtain a chromatic nonlinearity index that is roughly independent of the overall firing rate range of the cell.

**UV-green index.** To measure the relative response strength under pure UV versus green stimulation for each cell, we applied a UV-green index similar to previous work[42]. From the stimuli with pure color (±20% of green and ±20% of UV as part of the chromatic-integration stimulus), we determined the peak firing rate in the PSTHs (between 50 and 250 ms after stimulus onset, 10 ms bin size) for each color ($f_{green}$ and $f_{UV}$, respectively), independent of the contrast sign. We defined the UV-green index as the normalized difference between the responses to the two colors, UV-green index $= \left( f_{green} - f_{UV} \right) / \left( f_{green} + f_{UV} \right)$. The index ranges between ±1, with positive values indicating stronger responses for green contrast and negative values indicating stronger responses for UV contrast.

**UV-selective and color-opponent cells.** The UV-green index was used to identify UV-selective cells as those cells with UV-green index smaller than −0.7 (Supplementary Fig. 2a, c). Furthermore, cells with no crossing point of the chromatic-integration curves that were not labeled as UV-selective were considered as color-opponent cells (Supplementary Fig. 2b, c).

**Relative UV sensitivity at the balance point.** To quantify the sensitivity to UV versus green light for each ganglion cell at the balance point, we computed the relative UV sensitivity from the contrast values at the crossing point of the chromatic-integration curves. The crossing was identified as above for the computation of the chromatic nonlinear index, and the estimate of the contrast values was then obtained through linear interpolation between the four data points that delimited the crossing point. From the absolute contrast values $C_{UV}$ and $C_{green}$ of UV and green light at the interpolated crossing point, we computed the relative UV sensitivity as $1 - C_{UV} / (C_{UV} + C_{green})$. This value can range from zero to unity, and larger values correspond to higher sensitivity to UV light (because less UV contrast is required to balance the green contrast), whereas smaller values imply higher sensitivity to green light.

**On–Off index.** To classify ganglion cells as On, Off, or On–Off cells, we defined an On–Off index. From the stimuli with pure color (±20% of green or ±20% of UV), we selected the color with the maximum response R (average firing rate from 50 to 250 ms after stimulus onset with baseline rate 200 ms before the onset subtracted). From responses to this color, we calculated a normalized On–Off index as the difference of the responses to the On (+20%, $R_{On}$) and Off (−20%, $R_{Off}$) presentation, divided by the sum of the absolute values of the two responses, On-Off index $= \left( R_{On} - R_{Off} \right) / \left( \left| R_{On} \right| + \left| R_{Off} \right| \right)$. This index has a range of ±1, and we used a threshold of ±0.6 to separate Off and On cells from On–Off cells.

**Balance point polarity bias.** To check if the nonlinearity of nonlinear On–Off cells is driven by the Off or the On responses at the balance point, we defined a balance point polarity bias. We measured the slope of the green-On-UV-Off curve at the balance point as the difference between the firing rates of the two data points around the crossing point and subtracted it from the corresponding slope of the green-Off-UV-On curve (Supplementary Fig. 3a). Positive values indicate that responses at the balance point are driven by On stimulus components, whereas negative values indicate that Off stimulus components are more relevant.

**Chromatic grating stimulus.** The chromatic grating stimulus was a slowly moving UV-green sinusoid with opposite spatial phases of the two colors. We used 11 different contrast combinations, matching the contrast combinations of the spatially homogeneous chromatic-integration stimulus. The spatial period of the grating was set to 480 μm, and the temporal period was 1 s. Each contrast combination was presented for 10 s continuously before the next UV-green combination was selected randomly (using Fisher–Yates random permutation algorithm) with no intervals between the stimuli. The stimulus was presented for 30–60 repetitions per contrast combination. For each ganglion cell and each contrast combination, a PSTH (10-ms bins) over one temporal period was constructed from all repeats, leaving out the first temporal period after each switch in contrast. Using Fourier analysis, we then computed the amplitude at the first harmonic (F1 at 1 Hz, the frequency of stimulation) and at the second harmonic (F2 at 2 Hz, twice the frequency of stimulation). We defined a grating nonlinearity index as the ratio of the second-harmonic amplitude divided by the first-harmonic amplitude, evaluated at the contrast combination with the smallest first harmonic. This contrast combination was chosen because a small F1 component implies being near the balance point. The index only takes positive values, and large values (typically values larger than unity in analogy to analyses of spatial integration) indicate chromatic nonlinearity. The definition of this index and a threshold of unity are similar to studies of spatial integration with reversing gratings[12,50]. Cells with a maximum firing rate smaller than 5 Hz across all PSTHs from the different contrast combinations were excluded from this analysis, as evaluation of harmonics became unreliable at low firing rates (206 excluded out of 2167 cells).

**Local chromatic-integration stimulus.** We used a spatially local version of the chromatic-integration stimulus to probe for the involvement of the receptive field surround in nonlinear chromatic integration. The strategy was similar to previous studies of the effects of receptive field surround in mouse visual cortex[85,86]. In our case, the stimulus aimed at presenting the same contrast combinations of UV and green light as used in the full-field chromatic-integration stimulus, but spatially restricted to small regions roughly inside individual ganglion cell receptive fields. To make this stimulus efficient while covering the entire recording area, multiple randomly selected locations were stimulated simultaneously. Concretely, the stimulus was structured into step-like contrast presentations of 500 ms duration with no interval between successive presentations. For every 500-ms presentation, stimulus locations were randomly selected from a square grid (44 × 24 vertices, 160 μm between neighboring vertices) in a way so that, for every chosen vertex, 24 vertices in a square area around it (2 vertices or 320 μm in each of the four directions along the grid) were avoided. Centered on each selected vertex, a spot with a diameter of 160 μm was presented with a UV/green contrast combination, randomly chosen (using again Fisher–Yates randomization) from the same set of 22 combinations as used in the full-field chromatic-integration stimulus. The rest of the screen remained at background illumination. Note that the simultaneously presented contrast combinations at different locations were chosen independently of each other. On average, around 80-90 locations were selected for simultaneous display (until all vertices lay within the square areas of 5 × 5 vertices around selected locations), and each location was chosen on average every 6.4 ± 1.2 s (mean ± SD). We recorded responses under this stimulus for 65 minutes on average, which led to each location being chosen 595 ± 164 times, providing 27 ± 7 trials for each contrast combination.

For each cell, we constructed chromatic-integration curves for every location of the stimulus grid in the same way as for the full-field chromatic-integration stimulus (see above). For further analysis, such as computing the chromatic nonlinearity index, we selected the location for which the sum of the absolute values of the responses to the pure-color stimuli (±20% green and UV) was maximal. To check whether this location indeed matched the receptive field center of the cell, we measured the Euclidean distance between the selected location and the center point of the receptive field acquired under spatiotemporal white-noise stimulation (Fig. 6b). We excluded cells from the final analysis for which this distance was larger than the receptive field radius (122 excluded from 1219 cells) to ensure that the selected location indeed stimulated the receptive field center.

**Contrast reversing-grating stimulus.** We measured the spatial integration properties of ganglion cells using monochromatic square-wave grating stimuli (100% contrast) with spatial periods of 32, 64, 128, 224, and 448 μm, and we also included a spatially homogeneous full-field stimulus for comparison. For spatial periods of 64 and 128 μm, two different spatial phases (separated by 90°), and for spatial periods of 224 and 448 μm, four spatial phases (separated by 45°) were applied. The polarity of each grating was reversed every second for a total of 30

reversals. For each cell and each applied spatial period and phase, we constructed a PSTH over one temporal period (10-ms bins), excluding the first period after stimulus onset. The amplitudes of the first and second harmonics were computed by taking the Fourier transform of the PSTHs and extracting the amplitudes at the stimulus frequency and at twice the stimulus frequency. We then determined a spatial nonlinearity index by taking the maximum second-harmonic amplitude across all grating widths and phases divided by the maximum first-harmonic amplitude across all grating widths and phases.

**Models of chromatic integration for natural scenes.** For investigating linear and nonlinear chromatic integration with natural scenes, we constructed models of chromatically linear and nonlinear Off cells and simulated responses of populations of such cells to presentations of natural scenes with UV and green color channels. The UV and green images were taken from the "UV/Green Image Databases"[56,57] and show scenes of vegetation and sky, which were simultaneously recorded through a dichroic mirror with two cameras and color filters for green light (peak sensitivity at 500 nm, 70 nm bandwidth) and UV light (peak sensitivity at 350 nm, 50 nm bandwidth). Image resolution was $550 \times 300$ pixels. For use in our simulations, we separately converted each color channel in each image to Weber contrast $C = (I - I_B)/I_B$, where $I$ is the pixel intensity and $I_B$ is the mean intensity value of the corresponding color channel over the image. Using Weber contrast for each color channel individually assumes that the two types of opsins adapt independently to the mean of recently encountered light intensity in the corresponding color channel and that this light intensity history was sampled from the range covered by the image.

In order not to rely on specific assumptions about the mechanism behind nonlinear chromatic integration, we used a phenomenological, generic modeling framework. It applies a circular Gaussian of 25 pixels standard deviation as an Off-type receptive field, capturing the combined effect of center and surround. For reference, assuming that the image height corresponds to 40 meters when viewed from a distance of 15 meters, the 1-sigma diameter of the receptive field would correspond to 18 degrees of visual angle. Both the linear and the nonlinear chromatic-integration model filter the two chromatic input channels independently through this receptive field, and the nonlinear model then applies a half-wave rectification to each filtered signal. This rectification is the only difference between the two models and serves as a simple way to yield increased activity when UV and green light have opposing net contrast inside the receptive field, as observed in the recorded responses of chromatically nonlinear Off cells. In both models, the signals from the two chromatic channels are then summed and half-wave rectified to yield a firing rate output. For each image, the two models were evaluated on 10,000 randomly selected patches of $50 \times 50$ pixels (avoiding multiple selections of the same patch).

To measure the shift in the contrast of UV and green light when going from the sky region to the foreground region below, we fitted a sigmoid (logistic function) to the contrast values of each color channel along the pixels of each vertical column in an image. We compared the midpoints for the two color channels, excluding columns where the fit yielded a midpoint outside the range of the image.

**Detection of direction and orientation selectivity.** We identified direction-selective (DS) cells from their responses to drifting gratings (100% contrast) in eight equidistant directions. The gratings were either square-wave with 600 μm spatial period and 0.75 Hz temporal frequency[87] or sinusoidal with 250 μm spatial period and 0.6 Hz temporal frequency[88]. Each direction was presented for 6.67 s, separated by 3–5 s of background illumination. To identify orientation-selective (OS) cells, we used square-wave drifting gratings with 240 μm spatial period and 4 Hz temporal frequency[89] presented with eight equidistant directions (3 s per direction and 2 s gray screen between). All stimulus sequences were repeated four times.

We measured the direction tuning for each cell by calculating the mean firing rates $R_\theta$ in response to each direction $\theta (0 \le \theta < 2\pi)$, leaving out the onset response during the first period of the stimulus. We then computed a direction-selectivity index as the normalized vector sum:

$$\text{DSI} = \frac{\left| \sum_\theta R_\theta e^{i\theta} \right|}{\sum_\theta R_\theta}$$

In addition, we assessed the statistical significance of direction tuning, using a permutation test[47,90]. We created surrogate data by shuffling the spike counts randomly across all angles and repetitions of the stimulus 1000 times. For each shuffling, we calculated a DSI to generate a null distribution of DSIs and used the percentile of the true DSI as a p-value of direction tuning. As a criterion for direction selectivity, we required DSI > 0.3 with a p-value < 0.05. Cells with mean firing rates <1 Hz across different directions of the drifting grating stimuli were excluded from this analysis.

Analogously, to classify orientation-selective cells, we defined an orientation-selectivity index:

$$\text{DSI} = \frac{\left| \sum_\theta R_\theta e^{2i\theta} \right|}{\sum_\theta R_\theta}$$

Cells that were not classified as DS and that had OSI > 0.3 with a p-value < 0.05 from a permutation test and an average firing rate >1 Hz were considered as OS cells.

**Statistics and reproducibility.** Error measures denote standard error of the mean unless otherwise noted. The statistical significance of differences in baseline firing rates of cells with linear and nonlinear chromatic integration was assessed by a two-sided Kruskal–Wallis test, and the differences between individual groups were checked by performing a post-hoc analysis of mean ranks with Bonferroni correction. The statistical significance of the changes in the chromatic nonlinearity index of the cells between full-field and local stimulation and the changes before and after the administration of the pharmacological agents were assessed by a two-sided Wilcoxon signed-rank test.

The distributions of the nonlinearity indices measured before and after the application of the pharmacological agents or increases in the light level were represented as boxplots, which display the median by a central line and the interquartile range (IQR) from the first to the third quartile by a box. In addition, whiskers extend to the most extreme values within $1.5 \times$ IQR. The data points from which the boxplot was created are shown next to each boxplot.

To ensure reproducibility, all experiments were repeated in multiple retinas (31 total) obtained from different animals (19 total) with consistent results. Moreover, experimental conditions were optimized to guarantee data reproducibility and consistency between different experiments. Whenever possible, the order of stimuli and the order of presented contrasts were randomized to reduce the effects of light adaptation or biased measurement of the response of retinal ganglion cells.

**Reporting summary.** Further information on research design is available in the Nature Research Reporting Summary linked to this article.

## Data availability

The spike-time data that support the findings of this study are available at https://gin.g-node.org/gollischlab/Khani_and_Gollisch_2021_RGC_spike_trains_chromatic_integration.

## Code availability

The code to generate and visualize chromatic integration, chromatic grating, local chromatic integration, and spatiotemporal binary white-noise stimuli is available at https://gin.g-node.org/gollischlab/Khani_and_Gollisch_2021_RGC_spike_trains_chromatic_integration. The code used to analyze the chromatic-integration stimulus and create chromatic-integration curves is available at https://github.com/gollischlab/ChromaticIntegrationAnalysis. Images used to construct the models of chromatic integration for natural scenes were taken from the UV/Green Image Databases, available at https://www.ti.uni-bielefeld.de/html/people/ddiffert/databases_uvg.html.

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

# ARTICLE

75. Wallace, D. J. et al. Rats maintain an overhead binocular field at the expense of constant fusion. *Nature* **498**, 65–69 (2013).
76. Liu, J. K. et al. Inference of neuronal functional circuitry with spike-triggered non-negative matrix factorization. *Nat. Commun.* **8**, 149 (2017).
77. Shah, N. P. et al. Inference of nonlinear receptive field subunits with spike-triggered clustering. *Elife* **9**, e45743 (2020).
78. Heitman, A. et al. Testing pseudo-linear models of responses to natural scenes in primate retina. Preprint at https://www.biorxiv.org/content/10.1101/045336v2 (2016).
79. Pouzat, C., Mazor, O. & Laurent, G. Using noise signature to optimize spike-sorting and to assess neuronal classification quality. *J. Neurosci. Methods* **122**, 43–57 (2002).
80. Wei, W., Elstrott, J. & Feller, M. B. Two-photon targeted recording of GFP-expressing neurons for light responses and live-cell imaging in the mouse retina. *Nat. Protoc.* **5**, 1347–1352 (2010).
81. Field, G. D. & Rieke, F. Nonlinear signal transfer from mouse rods to bipolar cells and implications for visual sensitivity. *Neuron* **34**, 773–785 (2002).
82. Gauthier, J. L. et al. Receptive fields in primate retina are coordinated to sample visual space more uniformly. *PLoS Biol.* **7**, e1000063 (2009).
83. Wolfe, J. & Palmer, L. A. Temporal diversity in the lateral geniculate nucleus of cat. *Vis. Neurosci.* **15**, 653–675 (1998).
84. Fisher, R. A. & Yates, F. *Statistical Tables for Biological, Agricultural and Medical Research* (1948).
85. Hawrylycz, M. et al. Inferring cortical function in the mouse visual system through large-scale systems neuroscience. *Proc. Natl Acad. Sci. USA* **113**, 7337–7344 (2016).
86. de Vries, S. E. J. et al. A large-scale standardized physiological survey reveals functional organization of the mouse visual cortex. *Nat. Neurosci.* **23**, 138–151 (2020).
87. Kühn, N. K. & Gollisch, T. Joint encoding of object motion and motion direction in the salamander retina. *J. Neurosci.* **36**, 12203–12216 (2016).
88. Sabbah, S. et al. A retinal code for motion along the gravitational and body axes. *Nature* **546**, 492–497 (2017).
89. Vlasits, A. L. et al. Visual stimulation switches the polarity of excitatory input to starburst amacrine cells. *Neuron* **83**, 1172–1184 (2014).
90. Ecker, A. S. et al. State dependence of noise correlations in macaque primary visual cortex. *Neuron* **82**, 235–248 (2014).

## Acknowledgements
This work was supported by the Deutsche Forschungsgemeinschaft (DFG, German Research Foundation)—project IDs 154113120 (SFB 889, project C01) and 432680300 (SFB 1456, project B05)—and by the European Research Council (ERC) under the European Union's Horizon 2020 research and innovation program (grant agreement number 724822). We thank Daisuke Takeshita for his contribution to the initial part of the work, Michael Weick for his help in building the projection system, Dimokratis Karamanlis for assistance with data analysis, and Helene Schreyer for assistance with data analysis and comments on the draft of the manuscript.

## Author contributions
M.H.K. and T.G. conceived and designed experiments and data analysis methods. M.H. K. assembled the UV/green projection systems and the perforated-MEA recording set-ups, performed experiments, analyzed data, and prepared figures. M.H.K. and T.G. wrote the paper.

## Funding

## Competing interests
The authors declare no competing interests.
