## [Peer Review File · Nature Communications]

Reviewers' Comments:

Reviewer #1:

Remarks to the Author:

In this manuscript, Khani & Gollisch study how chromatic streams are integrated in the mouse retina. Mice, like most mammals, have two opsins. In mice, these chromatic streams carry signals from green and UV parts of the spectrum. However, apart from some previous work on color-opponency, the integration of these color streams in non-color-opponent cells remains unexplored. Drawing inspiration from the studies of spatial integration, they used a clever stimulus design to find the 'null' point where responses to these two streams would cancel each other if they were linearly superimposed. Yet, they find that for a significant number of cells, the responses are shifted from baseline; below baseline for ON cells and above baseline for OFF or ON-OFF cells. This non-linear effect was shown to depend on the participation of the surround and was diminished by blocking inhibitory neurotransmitters. This chromatic non-linearity was also shown to be largely independent of nonlinearities in spatial integration.

Although the experiments and data presented are sound and well-executed, there seems to be a fundamental gap in the interpretation of the data.

Major concern: The two streams of chromatic information are speculated to reach ganglion cells from cone-specific inputs. Thus, the combination of Green and UV responses is assumed to be integrated somehow across the entire retina. The problem with the former hypothesis is that it contradicts the inhomogeneous cone distribution in the mouse retina. While independent cone channels are present in the dorsal retina (via a very small number of true S-cones and M-opsin-expressing M-cones), cones in the ventral retina almost exclusively respond only to UV wavelengths [1-3]. Thus, the non-linear chromatic integrators (observed in the ventral retina) could not be getting their inputs from two different cone pathways. Importantly, for this argument's sake, non-linear chromatic integration can only occur if there are different chromatic channels. A more plausible explanation is that this non-linear effect is a result of the integration of signals from the rod pathway and that from S-cones [3-4]. Such an underlying circuit would also explain why the integration is mostly linear in the dorsal retina; the excitation profiles of M-opsin and rhodopsin are for in this study likely indistinguishable. Thus, linearity has to be assumed dorsally.

Unfortunately, the origins of these two chromatic channels have been largely left as an afterthought, rather than informing the choice of experiments. This should not only be discussed throughout the manuscript but also explored experimentally. In particular, since rods are known to be responsive at the light levels used in the experiments [5] and that their response speed is dependent on the used light intensity [6]. It would be interesting to see if non-linear and linear cells remain non-linear and linear, respectively, at varying light levels. These results could indicate that the measured responses are not a particularity of the measured light intensity, but a feature of the retina circuitry. Thus, I would suggest exploring the relationship of the light level by increasing/decreasing the mean luminance by one order of magnitude, perhaps reducing it even more.

Besides, given previous results [3,4,7], it appears that disrupting the horizontal cell in the outer retina would be an ideal experiment to dissect the pathway specifically. Horizontal cells have been shown to convey rod signals to cones (and vice versa, too), giving a greener surround to most of the cells.

Minor: There seems to be a lack of statistical power in the non-linear ON cells in fig 7e; even the example cell in fig 7b seems to have been linearized by the drug. Distribution of UV sensitive cells in Fig 8c is trivial since they were used to orient the retina in the first place.

1. Baden, T. et al. A Tale of Two Retinal Domains: Near-Optimal Sampling of Achromatic Contrasts in Natural Scenes through Asymmetric Photoreceptor Distribution. *Neuron* (2013).
2. Chang et al. Chromatic coding from cone-type unselective circuits in the mouse retina. *Neuron* (2013).

3. Szatko et al. Neural circuits in the mouse retina support color vision in the upper visual field. *BioRxiv* (2019)
4. Joesch, M. & Meister, M. A neuronal circuit for colour vision based on rod–cone opponency. *Nature* (2016).
5. Soucy, E., Wang, Y., Nirenberg, S., Nathans, J. & Meister, M. A Novel signaling pathway from rod photoreceptors to ganglion cells in mammalian retina. *Neuron* (1998).
6. Nikonov, S. S., et al. Physiological features of the S- and M-cone photoreceptors of wild-type mice from single-cell recordings. *J. Gen. Physiol.* (2006)
7. Szikra, T. et al. Rods in daylight act as relay cells for cone-driven horizontal cell-mediated surround inhibition. *Nature Neurosci.* (2014)

Reviewer #2:

Remarks to the Author:

In the vertebrate retina, spiking responses of different types of retinal ganglion cells are thought to represent distinct features in the visual input. Extensive research has been focusing on the underlying retinal circuits that integrate achromatic visual signals over space and time. However, chromatic signal integration has only been studied in selected ganglion cell types that exhibit color-selective or color-opponent properties. This study by Khani and Gollisch investigates chromatic integration in the mouse retinal ganglion cell population using multielectrode array recordings. Analogous to classical studies of spatial integration, they designed a novel set of chromatic stimuli with opposing contrasts of UV and green lights to study how signals from the two chromatic channels collectively shape ganglion cell spiking. They reported nonlinear chromatic integration in subsets of On, Off and On-Off retinal ganglion cells in the mouse retina in addition to the existing color-selective or color-opponent ganglion cell types. Moreover, they showed that the nonlinear chromatic integration was linearized during grating stimuli, local stimuli or with the application of inhibitory synaptic blockers, indicating the involvement of inhibitory inputs from the receptive field surround underlying the nonlinear chromatic integration. Overall, this is an elegant study with careful experimental design and data analysis, addressing an important gap in the understanding of the retinal code. The manuscript is beautifully written. The biological relevance of this nonlinear chromatic integration is explored and speculated.

Minor comment:

The authors used the wording “suppressive surround” to describe the putative mechanism of nonlinear chromatic integration. However, in the examples in Fig. 6 and Fig. S3, the firing rates during full field stimuli were not noticeably lower than those during local stimuli, and were even higher in some examples. So the surround mechanism involved in chromatic integration may not be exclusively suppressive, at least according to the conventional definition of suppressing RGC center response. It would be helpful to clarify this point.

Wei Wei
University of Chicago

Reviewer #3:

Remarks to the Author:

In this study Khani and Gollisch have explored an important and novel feature of colour processing by retinal output neurons in the mouse retina using multi-electrode array recordings together with carefully constructed chromatic light stimuli and modeling. Although it is well established that retinal ganglion cells nonlinearly integrate visual inputs over time and space, it is not known whether they do so for color. This is particularly interesting in the context of the cone opsin gradient present in the mouse retina with more medium (M) wavelength sensitive opsin expression in the dorsal retina and short (S) wavelength sensitive opsin expression in the ventral retina. By sampling over 2000 retinal

ganglion cells (RGCs) and carefully titrating the chromatic strength of S and M opsin activation, Khani and Gollisch have identified that about a third of the RGCs (in their sample size) exhibit nonlinear chromatic integration. They have further analyzed if there is correlation of chromatic and spatial integration by RGCs and what could be a possible circuit mechanism behind this nonlinear chromatic integration. They also analyzed the topographic distribution of RGCs across the mouse retina based on their chromatic integration properties. Interestingly they find that a greater fraction of RGCs that exhibit nonlinear chromatic integration are present in the ventral compared to the dorsal retina. The authors further show that for a number of natural scene images a nonlinear chromatic model of RGC signal integration performed better than a linear model. This is a beautiful and rigorous study identifying several novel features of chromatic processing in RGCs. The experiments are well thought-out and the data is of high quality. The manuscript is well-written and the analysis is thorough. I have a few questions/comments that might add to this already compelling study.

1. The authors limit the contrast of the S and M opsin stimulation to 20%. Does increasing the contrast beyond 20% increase the fraction of detected non-linear RGCs from 30% to higher? In other words does the balance point of nonlinear cells change with varying contrasts?
2. The authors could clarify how the responses of the ON-OFF nonlinear cells were calculated, was the spike rate calculated for both the ON and OFF phase of the response? Could they analyze the ON and OFF parts of the response separately? It will be interesting to know if in the ON-OFF nonlinear cells, which component has the stronger nonlinearity.
3. For nonlinear spatial integration of RGCs, bipolar cells form the nonlinear subunits in the RGC receptive field. Similarly is it possible to delineate the nonlinear chromatic subunits?
4. Could the authors attribute the chromatically nonlinear RGCs to some specific subtypes? Is there sampling of RGCs biased to GCs with larger soma and hence they dominate the population?
5. The pharmacology experiments with the inhibitory receptor blockers are always difficult to interpret given the non-specificity of action. However, there are some interesting effects for distinct nonlinear RGC type with specific drugs. Since GABA_A receptors are only localized to bipolar cell axon terminals in the mouse retina and thus mediate presynaptic inhibition, the effects of TPMPA is interesting since only the nonlinear ON cells are affected and not the OFF or ON-OFF cells. Moreover, only GABAzine linearizes the nonlinear OFF cells. In future the inhibitory mechanism underlying the nonlinear chromatic integration could be further dissected using genetic perturbations and single cell recordings. As a control, could the authors repeat the pharmacology with a localized spot stimulus where the surround is not activated. This should have minimal to no effect on the nonlinearity chromatic index.
6. In Suppl Fig 4 panel b, there seems to be good deal of variability in the fraction of nonlinear cells in dorsal vs ventral retina. It almost seems that the left and middle retina have an equal proportion of nonlinear cells in the dorsal and ventral retina. Could the authors elaborate on this?

We would like to thank the three reviewers for their efforts and the constructive feedback. In the revised manuscript, we have addressed the points raised by the reviewers as detailed in the responses below. The corresponding changes in the manuscript are marked in red.

Reviewer #1 (Remarks to the Author):

In this manuscript, Khani & Gollisch study how chromatic streams are integrated in the mouse retina. Mice, like most mammals, have two opsins. In mice, these chromatic streams carry signals from green and UV parts of the spectrum. However, apart from some previous work on color-opponency, the integration of these color streams in non-color-opponent cells remains unexplored. Drawing inspiration from the studies of spatial integration, they used a clever stimulus design to find the 'null' point where responses to these two streams would cancel each other if they were linearly superimposed. Yet, they find that for a significant number of cells, the responses are shifted from baseline; below baseline for ON cells and above baseline for OFF or ON-OFF cells. This non-linear effect was shown to depend on the participation of the surround and was diminished by blocking inhibitory neurotransmitters. This chromatic non-linearity was also shown to be largely independent of nonlinearities in spatial integration.

Although the experiments and data presented are sound and well-executed, there seems to be a fundamental gap in the interpretation of the data.

Major concern: The two streams of chromatic information are speculated to reach ganglion cells from cone-specific inputs. Thus, the combination of Green and UV responses is assumed to be integrated somehow across the entire retina. The problem with the former hypothesis is that it contradicts the inhomogeneous cone distribution in the mouse retina. While independent cone channels are present in the dorsal retina (via a very small number of true S-cones and M-opsin-expressing M-cones), cones in the ventral retina almost exclusively respond only to UV wavelengths [1-3]. Thus, the non-linear chromatic integrators (observed in the ventral retina) could not be getting their inputs from two different cone pathways. Importantly, for this argument's sake, non-linear chromatic integration can only occur if there are different chromatic channels. A more plausible explanation is that this non-linear effect is a result of the integration of signals from the rod pathway and that from S-cones [3-4]. Such an underlying circuit would also explain why the integration is mostly linear in the dorsal retina; the excitation profiles of M-opsin and rhodopsin are for in this study likely indistinguishable. Thus, linearity has to be assumed dorsally. Unfortunately, the origins of these two chromatic channels have been largely left as an afterthought, rather than informing the choice of experiments. This should not only be discussed throughout the manuscript but also explored experimentally. In particular, since rods are known to be responsive at the light levels used in the experiments [5] and that their response speed is dependent on the used light intensity [6]. It would be interesting to see if non-linear and linear cells remain non-linear and linear, respectively, at varying light levels. These results could indicate that the measured responses are not a particularity of the measured light intensity, but a feature of the retina circuitry. Thus, I would suggest exploring the relationship of the light level by increasing/decreasing the mean luminance by one order of magnitude, perhaps reducing it even more.

We thank the reviewer for these detailed suggestions. We agree that the previous version of the manuscript might not have made the putative contributions from rods clear enough. Following the reviewer's suggestions, we now include the potential role of rods in chromatic integration already at the outset of the study. Moreover, we performed new experiments with varying light intensities, as suggested, and found that increased light intensity indeed often linearizes chromatic integration. This confirms the interpretation that rod inputs are critical for nonlinear chromatic integration. The results are now shown in the new Figure 9a-b and form an integral part of the extended mechanistic interpretation and discussion (e.g. lines 656-681 in the Discussion).

Besides, given previous results [3,4,7], it appears that disrupting the horizontal cell in the outer retina would be an ideal experiment to dissect the pathway specifically. Horizontal cells have been shown to convey rod signals to cones (and vice versa, too), giving a greener surround to most of the cells.

Thanks for this suggestion. We have performed new experiments, in which we interfered with horizontal feedback by applying the pH buffer HEPES. Interestingly, this reduced the nonlinearity in Off and On-Off cells, but did not affect nonlinear On cells. These results are now shown in the new Figure 9c-d. We conclude that rod-cone interactions through horizontal cells do not provide a general mechanism for chromatic nonlinearities, but may contribute to shaping the nonlinearity in the (more complex) pathway of Off and On-Off cells (see extended discussion, e.g., lines 676-681 and 698-701).

Minor: There seems to be a lack of statistical power in the non-linear ON cells in fig 7e; even the example cell in fig 7b seems to have been linearized by the drug. Distribution of UV sensitive cells in Fig 8c is trivial since they were used to orient the retina in the first place.

To increase the statistical power in Fig. 7e, we performed additional experiments with TPMPA, more than doubling the number of nonlinear On cells here (lines 439-443). The new data show that TPMPA linearizes chromatically nonlinear On cells, confirming the previously seen trend. Regarding the distribution of UV-sensitive cells in Fig. 8c: indeed, this was not meant to be a finding, but is shown simply for comparison with the other distributions and as an illustration of our method of identifying the ventral side of the retina. This is now clarified in the figure legend.

- 1. Baden, T. et al. A Tale of Two Retinal Domains: Near-Optimal Sampling of Achromatic Contrasts in Natural Scenes through Asymmetric Photoreceptor Distribution. *Neuron* (2013).*
- 2. Chang et al. Chromatic coding from cone-type unselective circuits in the mouse retina. *Neuron* (2013).*
- 3. Szatko et al. Neural circuits in the mouse retina support color vision in the upper visual field. *BioRxiv* (2019)*
- 4. Joesch, M. & Meister, M. A neuronal circuit for colour vision based on rod–cone opponency. *Nature* (2016).*
- 5. Soucy, E., Wang, Y., Nirenberg, S., Nathans, J. & Meister, M. A Novel signaling pathway from rod photoreceptors to ganglion cells in mammalian retina. *Neuron* (1998).*
- 6. Nikonov, S. S., et al. Physiological features of the S- and M-cone photoreceptors of wild-type mice from single-cell recordings. *J. Gen. Physiol.* (2006)*
- 7. Szikra, T. et al. Rods in daylight act as relay cells for cone-driven horizontal cell-mediated surround inhibition. *Nature Neurosci.* (2014)*

Reviewer #2 (Remarks to the Author):

In the vertebrate retina, spiking responses of different types of retinal ganglion cells are thought to represent distinct features in the visual input. Extensive research has been focusing on the underlying retinal circuits that integrate achromatic visual signals over space and time. However, chromatic signal integration has only been studied in selected ganglion cell types that exhibit color-selective or color-opponent properties. This study by Khani and Gollisch investigates chromatic integration in the mouse retinal ganglion cell population using multielectrode array recordings. Analogous to classical studies of spatial integration, they designed a novel set of chromatic stimuli with opposing contrasts of UV and green lights to study how signals from the two chromatic channels collectively shape ganglion cell spiking. They reported nonlinear chromatic integration in subsets of On, Off and On-Off retinal ganglion cells in the mouse retina in addition to the existing

color-selective or color-opponent ganglion cell types. Moreover, they showed that the nonlinear chromatic integration was linearized during grating stimuli, local stimuli or with the application of inhibitory synaptic blockers, indicating the involvement of inhibitory inputs from the receptive field surround underlying the nonlinear chromatic integration. Overall, this is an elegant study with careful experimental design and data analysis, addressing an important gap in the understanding of the retinal code. The manuscript is beautifully written. The biological relevance of this nonlinear chromatic integration is explored and speculated.

Minor comment:

The authors used the wording “suppressive surround” to describe the putative mechanism of nonlinear chromatic integration. However, in the examples in Fig. 6 and Fig. S3, the firing rates during full field stimuli were not noticeably lower than those during local stimuli, and were even higher in some examples. So the surround mechanism involved in chromatic integration may not be exclusively suppressive, at least according to the conventional definition of suppressing RGC center response. It would be helpful to clarify this point.

Wei Wei

University of Chicago

Thanks for pointing this out, as it allows us to clarify this crucial aspect. In fact, it is difficult to interpret the change in firing rate between the global and local flashes. Local flashes evoke little surround activation, which should increase responses for cells with a suppressive surround. But local flashes often do not perfectly fill the receptive field center and can thereby lead to reduced center excitation and thus reduced responses. We now explain this when presenting the data from locally sparse stimulation (lines 381-388). The potentially reduced center excitation from not covering the entire receptive field center does not affect the interpretation of our results, as chromatic integration is also relevant within partially activated receptive fields. In addition, in the Discussion, we point out the possibility that an excitatory component could play a role in nonlinear chromatic integration, in particular for Off and On-Off cells (lines 688-691).

Reviewer #3 (Remarks to the Author):

In this study Khani and Gollisch have explored an important and novel feature of colour processing by retinal output neurons in the mouse retina using multi-electrode array recordings together with carefully constructed chromatic light stimuli and modeling. Although it is well established that retinal ganglion cells nonlinearly integrate visual inputs over time and space, it is not known whether they do so for color. This is particularly interesting in the context of the cone opsin gradient present in the mouse retina with more medium (M) wavelength sensitive opsin expression in the dorsal retina and short (S) wavelength sensitive opsin expression in the ventral retina. By sampling over 2000 retinal ganglion cells (RGCs) and carefully titrating the chromatic strength of S and M opsin activation, Khani and Gollisch have identified that about a third of the RGCs (in their sample size) exhibit nonlinear chromatic integration. They have further analyzed if there is correlation of chromatic and spatial integration by RGCs and what could be a possible circuit mechanism behind this nonlinear chromatic integration. They also analyzed the topographic distribution of RGCs across the mouse retina based on their chromatic integration properties. Interestingly they find that a greater fraction of RGCs that exhibit nonlinear chromatic integration are present in the ventral compared to the dorsal retina. The authors further show that for a number of natural scene images a nonlinear chromatic model of RGC signal integration performed better than a linear model. This is a beautiful and rigorous study identifying several novel features of chromatic processing in RGCs. The experiments are well thought-out and the data is of high quality. The manuscript is well-written and the analysis is thorough. I have a few questions/comments that might add to this already compelling

study.

1. The authors limit the contrast of the S and M opsin stimulation to 20%. Does increasing the contrast beyond 20% increase the fraction of detected non-linear RGCs from 30% to higher? In other words does the balance point of nonlinear cells change with varying contrasts?

This is an interesting suggestion, and we did attempt a few experiments with increased contrast ranges (e.g. up to 40% contrast). However, although we also observed linear and nonlinear chromatic integration, quantifying the degree of nonlinearity turned out to be much more difficult for two reasons. First, the larger contrast range meant reduced resolution for identifying the crossing point, which made assessing the response at the crossing point more difficult. (This consideration was our original reason for restricting our measurements to a range of +/-20% contrast.) Second, we found that going to a higher contrast range made the measured chromatic integration curves much noisier, potentially because of increased adaptation and thus a dependence on the previous stimuli in the flash sequence. Thus, comparing different contrast ranges will require a new, adjusted stimulus design, with longer recordings that allow more sampling points in the chromatic integration curves and longer intervals between successive stimuli. As we do not consider the contrast dependence to be critical for our goal of demonstrating linear and nonlinear chromatic integration, we therefore decided to limit the present work to the original contrast range and explore the effect of contrast in a more systematic fashion in the future.

2. The authors could clarify how the responses of the ON-OFF nonlinear cells were calculated, was the spike rate calculated for both the ON and OFF phase of the response? Could they analyze the ON and OFF parts of the response separately? It will be interesting to know if in the ON-OFF nonlinear cells, which component has the stronger nonlinearity.

Thanks for this interesting suggestion. Although we do not have a way to assess nonlinearities separately for On and Off stimulus components (as detection of nonlinearities is based on simultaneously presenting On and Off stimuli with different colors), we found a way to determine whether an observed nonlinearity was driven by On or by Off responses. The new analysis is now shown in the new Supplementary Fig. 3 (see also lines 252-267 in Results and lines 945-952 in Methods). Briefly, we measure whether, at the crossing point of the chromatic integration curves, responses increase towards the On-sides or towards the Off-sides of the curves. (Roughly speaking, each curve is typically U-shaped, with one side corresponding more to On responses and the other more to Off responses.) We found that chromatically nonlinear On-Off cells mostly show increased activity with increased Off-type stimulation at this point (or, in other words, that the crossing point is located on the Off-response sides of the curves), indicating that activity at the balance point is driven by Off responses. This indicates that nonlinear On-Off cells could become nonlinear through the same circuitry as nonlinear Off cells. This finding is in line with the observation that both these cell classes show increased activity at the balance point, unlike the decreased activity seen in nonlinear On cells.

3. For nonlinear spatial integration of RGCs, bipolar cells form the nonlinear subunits in the RGC receptive field. Similarly is it possible to delineate the nonlinear chromatic subunits?

The question whether the subunit idea from spatial integration can be transferred to chromatic integration is quite interesting conceptually. Subunits should correspond to circuit elements that integrate linearly over different ranges of the light spectrum and whose signals are then nonlinearly combined. Unlike for spatial integration, however, the nonlinear circuitry

seems to be more complicated, in particular as inhibitory signaling (and presumably serial inhibition) is involved, precluding a simple additive subunit model. How such suppressively interacting chromatic subunits could be identified is not immediately clear. We therefore cannot answer this question, but have taken it up in the Discussion when referring to challenges for including chromatic integration in computational models (lines 718-724).

4. Could the authors attribute the chromatically nonlinear RGCs to some specific subtypes? Is there sampling of RGCs biased to GCs with larger soma and hence they dominate the population?

Identifying subtypes of ganglion cells from multielectrode-array recordings presents still a considerable challenge. However, in the revised manuscript, we approach this challenge by analyzing two readily identifiable ganglion cell classes, direction-selective (DS) and orientation-selective (OS) ganglion cells. We find that DS cells are generally chromatically linear, whereas OS cells can be subdivided into linear and nonlinear cells (which we hypothesize to correspond to different subtypes of OS cells). This supports the idea that the chromatic nonlinearity does depend on ganglion cell type. These new analyses are shown in the new Supplementary Fig. 6 (see also lines 629-639 in the Discussion). Also, there is almost certainly a recording bias in multielectrode-array recordings. We now mention this when presenting the population overview of linear and nonlinear cells (lines 223-225).

5. The pharmacology experiments with the inhibitory receptor blockers are always difficult to interpret given the non-specificity of action. However, there are some interesting effects for distinct nonlinear RGC type with specific drugs. Since GABA_A receptors are only localized to bipolar cell axon terminals in the mouse retina and thus mediate presynaptic inhibition, the effects of TPMPA is interesting since only the nonlinear ON cells are affected and not the OFF or ON-OFF cells. Moreover, only GABA_A linearizes the nonlinear OFF cells. In future the inhibitory mechanism underlying the nonlinear chromatic integration could be further dissected using genetic perturbations and single cell recordings. As a control, could the authors repeat the pharmacology with a localized spot stimulus where the surround is not activated. This should have minimal to no effect on the nonlinearity chromatic index.

We agree that the pharmacology results are intriguing, but difficult to interpret. Genetic perturbations and single-cell recordings may help elucidate this, although there seems to be no straightforward experiment that would directly reveal the role of GABA-C in chromatic integration. The presynaptic nature of GABA-C, for example, makes it hard to study the effect in patch-clamp recordings from ganglion cells. We therefore believe that these are rather interesting questions for future studies and beyond the scope of the present work.

The control experiment with pharmacology and localized spot stimulus is an interesting suggestion. However, a problem here is that the localized spot already by itself linearizes the responses in a similar fashion as the inhibition block. Thus, adding inhibition blockers under localized stimulation cannot be used to test whether the relevant inhibition acts through the surround, as there is no longer a nonlinear response component that could remain unaltered. For this reason (and because combining the long locally sparse stimulation with the need to repeat measurements before, during, and after drug application would be stretching the limits of typical recording durations), we have decided not to perform these experiments.

6. In Suppl Fig 4 panel b, there seems to be good deal of variability in the fraction of nonlinear cells in dorsal vs ventral retina. It almost seems that the left and middle retina have an equal proportion of nonlinear cells in the dorsal and ventral retina. Could the authors elaborate on this?

This is a good observation, and it prompted us to show more data in this figure (now Supplementary Fig. 5) by attaching the histograms of the distributions of the three distinguished cell classes along the dorsal-ventral axis. This shows that the sampling of cells along this axis can be uneven (as in the first column of Supplementary Fig. 5), so that also the dorsal retina shows a good number of nonlinear cells. What remains consistent, however, is that the relative proportion of nonlinear cells (compared to linear cells) is generally larger in the ventral retina. This is now also pointed out in the legend of this figure.

Reviewers' Comments:

Reviewer #1:

Remarks to the Author:

The authors did a great job of addressing my concerns. The revisions were well done and improved the paper substantially. Their results and perspectives can now be assessed within all current findings in the field. I have nothing substantial more to add. I'm happy with the current version and revisions and would advise publishing the paper. Happy Holidays.

Reviewer #2:

Remarks to the Author:

The authors have addressed my concerns. Congratulations on the nice study.
Wei Wei
U. of Chicago

Reviewer #3:

Remarks to the Author:

The authors have done an excellent job in addressing the concerns I raised. I have one pending clarification about one of the issues raised by reviewer 1 about the contribution of rods for nonlinear chromatic integration. The authors have used a background luminance of 3000 R*/sec (mesopic) which is quite bright and mouse rods are mostly saturated at this light level (Grimes et al 2018 eLife; Fig. 4a). It would be better to use a much lower background light level (~100 R*/sec) to probe the full extent of chromatic nonlinearity and bolster the contribution of rods to this interesting property. Perhaps, if experiments are not feasible, then mentioning this limitation in the results/discussion would be beneficial for the readers and something that could be explored in further detail in future studies. Overall, this is an exciting study.

We would like to thank the three reviewers for their time and for the positive feedback.

Reviewer #1 (Remarks to the Author):

The authors did a great job of addressing my concerns. The revisions were well done and improved the paper substantially. Their results and perspectives can now be assessed within all current findings in the field. I have nothing substantial more to add. I'm happy with the current version and revisions and would advise publishing the paper. Happy Holidays.

Thank you!

Reviewer #2 (Remarks to the Author):

*The authors have addressed my concerns. Congratulations on the nice study.
Wei Wei
U. of Chicago*

Thank you!

Reviewer #3 (Remarks to the Author):

The authors have done an excellent job in addressing the concerns I raised. I have one pending clarification about one of the issues raised by reviewer 1 about the contribution of rods for nonlinear chromatic integration. The authors have used a background luminance of 3000 R/sec (mesopic) which is quite bright and mouse rods are mostly saturated at this light level (Grimes et al 2018 eLife; Fig. 4a). It would be better to use a much lower background light level (~100 R*/sec) to probe the full extent of chromatic nonlinearity and bolster the contribution of rods to this interesting property. Perhaps, if experiments are not feasible, then mentioning this limitation in the results/discussion would be beneficial for the readers and something that could be explored in further detail in future studies. Overall, this is an exciting study.*

We agree that comparison with lower background light level could be an interesting endeavor and could help further elucidate the integration of rod and cone signals. However, given that we do see clear evidence of rod contributions already at the applied higher light level and that we are currently not set up to perform experiments at a much lower light level, we cannot integrate such measurements in the present manuscript. As suggested, we mention this possibility in the Discussion.